

# When best is the enemy of good – critical evaluation of performance criteria in hydrological models

Guillaume Cinkus[1], Naomi Mazzilli[2], Hervé Jourde[1], Andreas Wunsch[3], Tanja Liesch[3], Nataša Ravbar[4], Zhao Chen[5], and Nico Goldscheider[3]

[1]HydroSciences Montpellier (HSM), Univ. Montpellier, CNRS, IRD, 34090 Montpellier, France
[2]UMR 1114 EMMAH (AU-INRAE), Université d'Avignon, 84000 Avignon, France
[3]Karlsruhe Institute of Technology (KIT), Institute of Applied Geosciences, Kaiserstr. 12, 76131 Karlsruhe, Germany
[4]ZRC SAZU, Karst Research Institute, Titov trg 2, 6230 Postojna, Slovenia
[5]Institute of Groundwater Management, Technical University of Dresden, 01062 Dresden, Germany

*Correspondence to:* Guillaume Cinkus (guillaume.cinkus@umontpellier.fr)

**Abstract.** Performance criteria play a key role in the calibration and evaluation of hydrological models and have been extensively developed and studied, but some of the most used criteria still have unknown pitfalls. This study set out to
examine counterbalancing errors, which are inherent to the Kling-Gupta Efficiency (KGE) and its variants. A total of nine performance criteria – including the KGE and its variants, as well as the Nash-Sutcliffe Efficiency (NSE) and the refined version of the Willmott's index of agreement ($d_r$) – were analysed using synthetic time series and a real case study. Results showed that, assessing a simulation, the score of the KGE and some of its variants can be increased by concurrent over- and underestimation of discharge. These counterbalancing errors may favour bias and variability parameters, therefore preserving
an overall high score of the performance criteria. As bias and variability parameters generally account for 2/3 of the weight in the equation of performance criteria such as the KGE, this can lead to an overall higher criterion score without being associated to an increase in model relevance. We recommend using (i) performance criteria that are not or less prone to counterbalancing errors (NSE, $d_r$, modified KGE, non-parametric KGE, Diagnostic Efficiency) in a multi-criteria framework, and/or (ii) scaling factors in the equation to reduce the influence of relative parameters.



## 1    Introduction

Hydrological models are fundamental to solve problems related to water resources. They help characterising hydrosystems (Hartmann et al., 2014), predicting floods (Kauffeldt et al., 2016; Jain et al., 2018) and managing water resources (Muleta and Nicklow, 2005). A lot of research efforts are thus dedicated to improve the reliability, the robustness and the relevance

of such models. Improvements can be made by working on (i) input data, (ii) model parameters and structure, (iii) uncertainty quantification, and also (iv) model calibration (Beven, 2019). In this study, we focus on the proper use of performance criteria for calibrating and evaluating hydrological models – an important part that can easily be overlooked (Jackson et al., 2019).

A performance criterion aims to evaluate the goodness-of-fit of a model to an observed data. It is generally expressed as a

score, for which the best value corresponds to a perfect fit between predictions and observations. In hydrology, the Nash-Sutcliffe Efficiency (NSE) (Nash and Sutcliffe, 1970) is still one of the most commonly used criteria (Kling et al., 2012), although the past decade has seen a gain in popularity of alternatives (Clark et al., 2021), e.g. the Kling-Gupta Efficiency (KGE) (Gupta et al., 2009). Many authors have pointed out the inherent limitations of using performance criteria, especially the fact that a single score metric cannot reflect all relevant hydrological aspects of a model (Gupta et al., 2009). The use of a

multi-criteria framework is thus often emphasised to quantify different aspects of a model (Clark et al., 2021; Moriasi et al., 2015; Gupta et al., 1998; Jackson et al., 2019; van Werkhoven et al., 2009; Knoben et al., 2019; Althoff and Rodrigues, 2021; Ritter and Muñoz-Carpena, 2013; Krause et al., 2005; Legates and McCabe Jr., 1999), alongside a scientific evaluation of the results (Biondi et al., 2012). Knoben et al. (2019), Althoff and Rodrigues (2021) and Clark et al. (2021) pointed out that modellers should carefully think about which aspects they consider the most important in their hydrological

model and how to evaluate them.

Performance criteria also have shortcomings at a distinctive level. A number of studies have identified several limitations of the NSE: (i) the contribution of the normalised bias depends of the discharge variability of the basin, (ii) discharge variability is inevitably underestimated because the NSE is maximised when the variability equals the correlation coefficient, which is always smaller than unity, and (iii) mean flow is not a meaningful benchmark for highly variable discharges (Gupta et al.,

2009; Willmott et al., 2012). The KGE aims to address these limitations but also has its own issues (Gupta et al., 2009). Santos et al. (2018) identified pitfalls when using the KGE with a prior logarithmic transformation of the discharge. Knoben et al. (2019) warned against directly comparing NSE and KGE scores as the KGE has no inherent benchmark. Ritter and Muñoz-Carpena (2013) and Clark et al. (2021) showed that NSE and KGE scores can be strongly influenced by few data points, resulting in substantial uncertainties on the predictions.

What is not fully addressed yet is the trade-off between individual components (Wöhling et al., 2013) and especially the impact of counterbalancing errors induced by bias and variability parameters, which are integrated in many performance criteria. While accurate bias and variability are desired aspects of hydrological models, sometimes good evaluations may accidentally result from negative and positive values cancelling each other (Jackson et al., 2019; Massmann et al., 2018).



This can be particularly detrimental to model calibration and evaluation, as it generates an increase in criterion score without
necessarily being associated to a better model relevance. Some performance criteria naturally address this problem by using
absolute or squared error values, but other criteria such as the KGE and its variants do not, as they use relative errors. The
aim of this study is to assess the extent to which criteria scores can be trusted for calibrating and evaluating hydrological
models when predictions have concurrent over- and underestimated values. The influence of counterbalancing errors is
evaluated on nine performance criteria including the NSE and the KGE. This selection is far being from exhaustive but
includes widely used and recently proposed KGE variants, as well as more traditional criteria such as the NSE or the refined
version of the Willmott's index of agreement ($d_r$) for comparison purpose. We first use synthetic time series to highlight the
counterbalancing errors mechanism. Second, we show how counterbalancing errors can impair the interpretation of
hydrological models in a real case study. Finally, we provide some recommendations about the use of scaling factors and the
choice of appropriate performance criteria to nullify or reduce the influence of counterbalancing errors.

## 2    Performance criteria

### 2.1    Parameters description

All the performance criteria considered in this study are based on the same or similar statistical indicators, which are first
described to avoid repetition.

We use $x_o(t)$ and $x_s(t)$ to refer to observed and simulated values of calibration variable $x$ at a specific time step $t$. $r$ and $r_s$
correspond to the Pearson and the Spearman rank correlation coefficients (Freedman et al., 2007), respectively.

$\beta$ is the ratio between the mean of simulated values $\mu_s$ and the mean of observed values $\mu_o$:

$$\beta = \frac{\mu_s}{\mu_o} \tag{1}$$

$\beta_n$ corresponds to the bias (mean error) normalised by the standard deviation of observed values $\sigma_o$:

$$\beta_n = \frac{\mu_s - \mu_o}{\sigma_o} \tag{2}$$

$\alpha$ is the ratio between the standard deviation of simulated values $\sigma_s$ and the standard deviation of observed values $\sigma_o$:

$$\alpha = \frac{\sigma_s}{\sigma_o} \tag{3}$$

$\gamma$ is the ratio between the coefficient of variation of simulated values ($CV_s = \sigma_s/\mu_s$) and the coefficient of variation of
observed values ($CV_o = \sigma_o/\mu_o$):

$$\gamma = \frac{CV_s}{CV_o} \tag{4}$$

$\overline{B_{rel}}$ and $|B_{area}|$ (Schwemmle et al., 2021) are based on the Flow Duration Curve (FDC). $B_{rel}(i)$ is defined as the relative
bias of the simulated and observed flow duration curves at the exceedance probability $i$:





$$B_{rel} = \frac{x_s(i) - x_o(i)}{x_o(i)} \tag{5}$$

where $x_s(i)$ and $x_o(i)$ correspond to the simulated and observed values of calibration variable at exceedance probability $i$. $\overline{B_{rel}}$ is the mean of $B_{rel}(i)$ when looking at $n$ observations:

$$\overline{B_{rel}} = \frac{1}{n} \sum_{i=0}^{i=1} B_{rel}(i) \tag{6}$$

$|B_{area}|$ is calculated as follows:

$$|B_{area}| = \int_0^1 |B_{res}(i)| \, di \tag{7}$$

with $B_{res}$ the residual bias:

$$B_{res} = B_{rel}(i) - \overline{B_{rel}} \tag{8}$$

$\alpha_{NP}$ (Pool et al., 2018) is also based on the FDC:

$$\alpha_{NP} = 1 - \frac{1}{2} \sum_{k=1}^{n} \left| \frac{x_s(I(k))}{n\mu_s} - \frac{x_o(J(k))}{n\mu_o} \right| \tag{9}$$

where $I(k)$ and $J(k)$ stand for the time steps of the $k^{th}$ largest discharge for the simulated and observed time series, respectively.

As $\beta$, $\beta_n$ and $\overline{B_{rel}}$ all represent the bias, they are therefore designed as "bias parameters" in this study.

## 2.2    Score calculation

A total of nine performance criteria are analysed in this study: the NSE, KGE, 2012-version of the KGE or modified KGE (KGE'), 2021-version of the KGE (KGE''), non-parametric KGE (KGE$_{NP}$), Diagnostic Efficiency (DE), Liu-Mean Efficiency (LME), Least-squares Combined Efficiency (LCE) and d$_r$. The value considered as the best score is equal to one

for all criteria, except for the DE, for which it is equal to zero.

The NSE (Nash and Sutcliffe, 1970) is a normalised variant of the Mean Squared Error (MSE) and compares a prediction to the observed mean of the target variable:

$$NSE = 1 - \frac{\sum(x_s(t) - x_o(t))^2}{\sum(x_o(t) - \mu_o)^2} \tag{10}$$

Gupta et al. (2009) algebraically decomposed the NSE into correlation, variability, and bias components:

$$NSE = 2\alpha r - \alpha^2 + \beta_n^2 \tag{11}$$

The Kling-Gupta Efficiency (KGE) was proposed by Gupta et al. (2009) as an alternative to the NSE. The optimal KGE

corresponds to the closest point of the three-dimensional Pareto front – of $\alpha$, $\beta$ and $r$ – to the ideal value of [1; 1; 1]:

$$KGE = 1 - \sqrt{(\alpha - 1)^2 + (\beta - 1)^2 + (r - 1)^2} \tag{12}$$


A modified Kling-Gupta Efficiency was proposed by Kling et al. (2012). The coefficient of variation is used instead of the standard deviation to ensure that bias and variability are not cross-correlated:

$$KGE' = 1 - \sqrt{(\gamma - 1)^2 + (\beta - 1)^2 + (r - 1)^2} \tag{13}$$

Tang et al. (2021) proposed another variant (KGE'') by using the normalised bias instead of $\beta$ to ensure that the score is not overly sensitive to mean values – $\mu_o$ or $\mu_s$ – close to zero (Santos et al., 2018; Tang et al., 2021):

$$KGE'' = 1 - \sqrt{(\alpha - 1)^2 + \beta_n^2 + (r - 1)^2} \tag{14}$$

Pool et al. (2018) cautioned against the implicit assumptions of the KGE – data linearity, data normality and absence of outliers – and proposed a non-parametric alternative (KGE$_{NP}$) for limiting their impact. The non-parametric form of the variability is calculated using the Flow Duration Curve (FDC) and the Spearman rank correlation coefficient is used instead of the Pearson correlation coefficient:

$$KGE_{NP} = 1 - \sqrt{(\alpha_{NP} - 1)^2 + (\beta - 1)^2 + (r_S - 1)^2} \tag{15}$$

In a similar way, Schwemmle et al. (2021) used FDC-based parameters to account for variability and bias in another KGE
variant: the Diagnostic Efficiency. This criterion is based on constant, dynamic and timing errors and aims to provide a stronger link to hydrological processes (Schwemmle et al., 2021):

$$DE = \sqrt{\overline{B_{rel}}^2 + |B_{area}|^2 + (r - 1)^2} \tag{16}$$

In this study, we used a Normalised Diagnostic Efficiency (DE') so that the best error score equals to one for facilitating the comparison with other performance criteria:

$$DE' = 1 - \sqrt{\overline{B_{rel}}^2 + |B_{area}|^2 + (r - 1)^2} \tag{17}$$

Liu (2020) proposed another alternative, the Liu-Mean Efficiency, to improve the simulation of extreme events. The LME
thus aims to address the underestimation of variability of the KGE, which is still a concern despite being not as severe as with the NSE (Gupta et al., 2009; Mizukami et al., 2019):

$$LME = 1 - \sqrt{(r\alpha - 1)^2 + (\beta - 1)^2} \tag{18}$$

Lee and Choi (2022) proposed the Least-squares Combined Efficiency to address the shortcomings of the LME identified by Choi (2022): (i) an infinite number of solutions for the maximum score, and (ii) a inclination to overestimate high flows and underestimate low flows. The LCE is based on the least-squares statistics combined from both-way regression lines $r\alpha$ and
$r/\alpha$:

$$LCE = 1 - \sqrt{(r\alpha - 1)^2 + (r/\alpha - 1)^2 + (\beta - 1)^2} \tag{19}$$

Willmott et al. (2012) proposed a refined version of Willmott's index of agreement, which aim to address the issues associated with the NSE (Jackson et al., 2019):



$$d_r = \begin{cases} 1 - \dfrac{\sum |x_s(t) - x_o(t)|}{2\sum |x_o(t) - \mu_o|} & when \quad \sum |x_s(t) - x_o(t)| \leq 2\sum |x_o(t) - \mu_o| \\ \dfrac{2\sum |x_o(t) - \mu_o|}{\sum |x_s(t) - x_o(t)|} - 1 & when \quad \sum |x_s(t) - x_o(t)| > 2\sum |x_o(t) - \mu_o| \end{cases} \tag{20}$$

## 3    Synthetic time series

### 3.1    Generating synthetic time series with homothetic transformations

A simulation performance can be assessed in terms of bias, variability and timing errors (Gupta et al., 2009). Bias and variability errors correspond to a difference in volume and amplitude of discharges. Timing errors correspond to a shift in time. We created a synthetic hydrograph corresponding to one flood event as the reference (observed) time series. We also generated synthetic transformations – of the reference time series – with different errors on bias and variability corresponding to time series simulated by a model. We did not consider any timing errors as our aim is to assess

counterbalancing errors induced by bias and variability parameters. Synthetic transformations were generated by multiplying the reference time series by a coefficient $\omega$:

$$Q_s(t) = Q_o(t) * \omega \tag{21}$$

where $Q_s(t)$ stands for the transformed discharge at the time $t$, $Q_o(t)$ the reference discharge at the time $t$ and $\omega$ a coefficient. $\omega$ values were sampled between -0.36 and 0.36 at a defined interval of 0.002 on a logarithmic scale to ensure a fair distribution between underestimated ($\omega < 1$) and overestimated ($\omega > 1$) transformations. This results in 361

transformations evenly distributed around the $\omega = 1$ homothety, which corresponds to the reference time series (i.e. absence of transformation). We defined $\omega$ bounds such that the transformed peak discharge roughly ranges from half ($\omega = 0.437$) to twice ($\omega = 2.291$) compared to the reference time series. Note that $\omega$ homotheties still induce small timing errors – which were considered negligible – because the correlation coefficients ($r$ and $r_s$) also slightly account for the shape of the transformation.





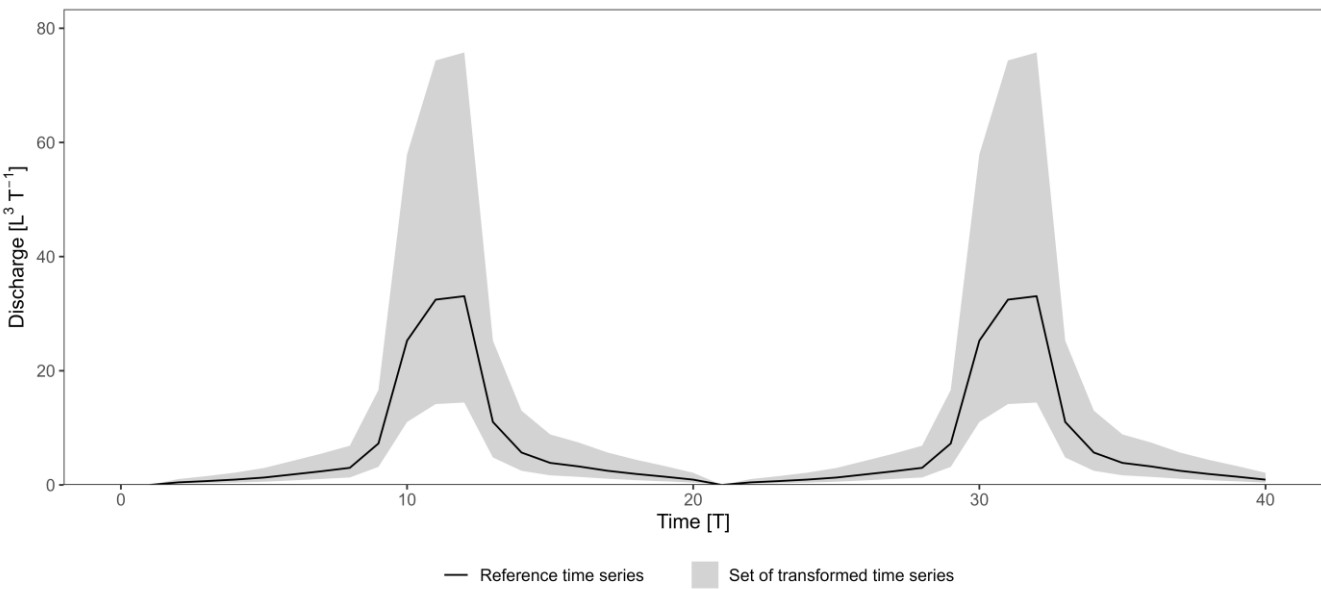


**Figure 1: Synthetic hydrograph corresponding to two flood events.**

To study counterbalancing errors induced by bias and variability parameters, we generated time series that consist of two successive flood events and considered all possible combinations of the 361 transformations for the simulated time series (Fig. 1). This results in a total of $361^2 = 130321$ transformations with two flood events, including (i) a "perfect" transformation with $\omega = 1$ for both flood events, (ii) "Bad-Good" (BG) or "Good-Bad" (GB) transformations when $\omega = 1$ for only one out of the two flood events, and (iii) "Bad-Bad" (BB) transformations when $\omega \neq 1$ for both flood events. The performance of the transformations – with regards to the reference time series – were evaluated using the nine performance criteria presented in Sect. 2.

**3.2    Identifying counterbalancing errors on a straightforward example**

Figure 2 presents two hydrographs extracted from the set of transformations: (i) a BB model with the combination $[\omega_1 = 0.75; \omega_2 = 1.2]$, and (ii) a BG model with the combination $[\omega_1 = 0.75; \omega_2 = 1]$. The BG model stands as a better model because it perfectly reproduces the second flood event and is identical to the BB model on the first flood ($\omega_1 = 0.75$). Nevertheless, the KGE and its variants – KGE', KGE'', $KGE_{NP}$, DE', LME and LCE – all favour the BB model, whereas only the NSE and the $d_r$ evaluate the BG model as better (Fig. 3a).





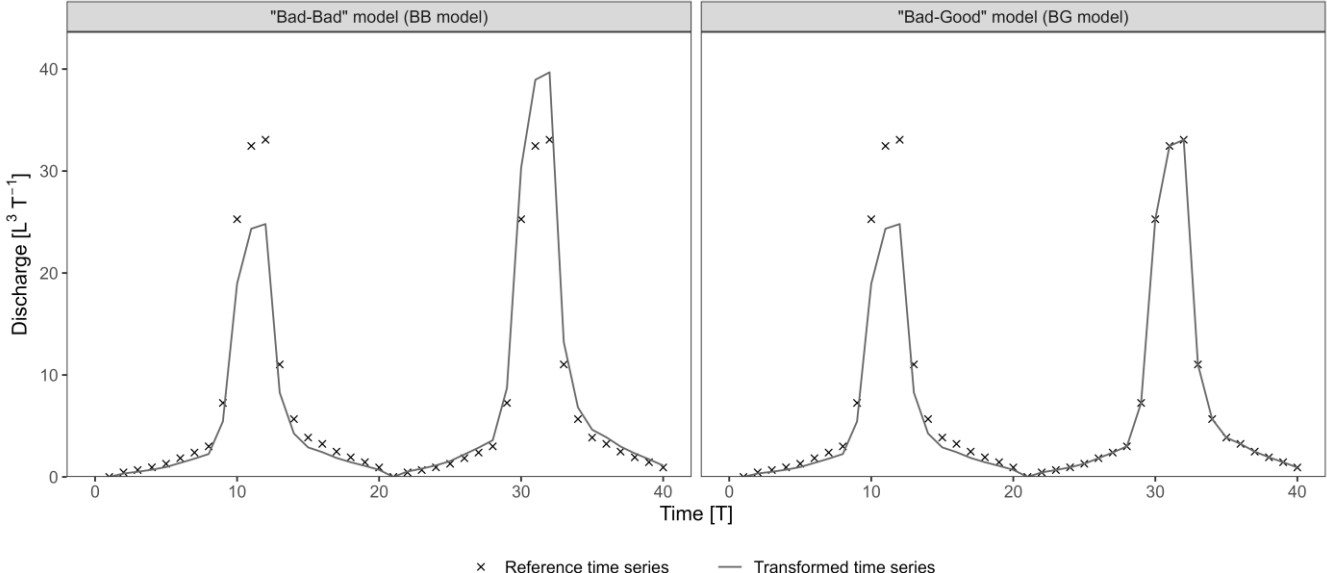

**Figure 2: Synthetic examples extracted from the set of transformations. The first and second flood events of the "Bad-Bad" and "Bad-Good" transformations were shifted with $[\omega_1 = 0.75; \omega_2 = 1.2]$ and $[\omega_1 = 0.75; \omega_2 = 1]$ combinations, respectively.**

The investigation of the components of the criteria (Fig. 3b) reveals how a seemingly better model (i.e. the BG model) can have a lower score than expected. Bias parameters are systematically better for the BB model, with 0.98 over 0.88 for $\beta$, -0.02 over -0.08 for $\beta_n$ and -0.04 over -0.12 for $\overline{Brel}$. Timing parameters are systematically better for the BG model, with 0.99 over 0.96 for $r$ and 0.99 over 0.98 for $r_s$. Variability parameters are mixed: (i) $\alpha$ favours the BB model with 1.01 over 0.89, (ii) $\gamma$ favours the BG model with 1.01 over 1.04, (iii) $\alpha_{NP}$ slightly favours the BG model with 0.94 over 0.93, and (iv)

$|B_{area}|$ is equal for both models. $r\alpha$ and $r/\alpha$ parameters are better for the BB model. $2\alpha r$ is better for the BG model.

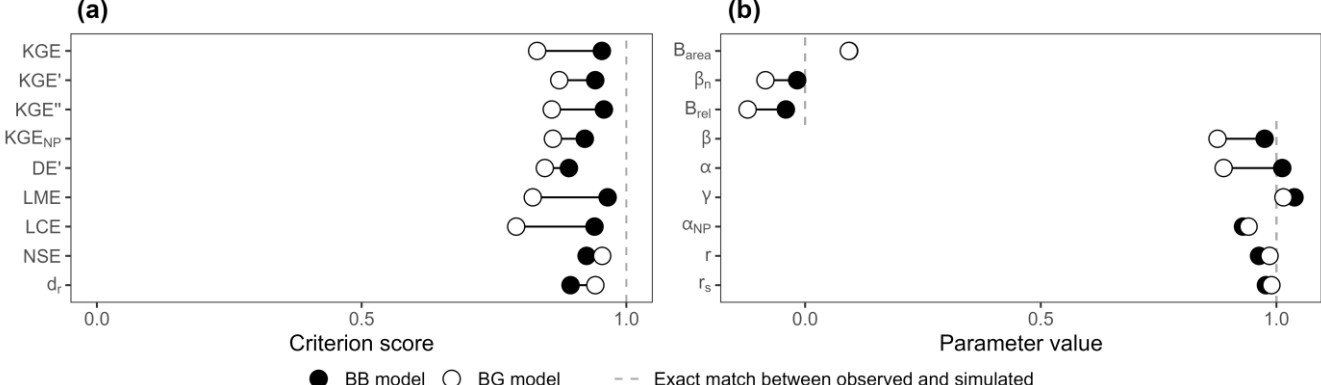

**Figure 3: (a) Score of the BB and BG transformations according to the different performance criteria. (b) Values of the parameters used in the calculation of the performance criteria.**



$\beta, \beta_n, \overline{Brel}, \alpha, r\alpha$ and $r/\alpha$ parameters all provide a better evaluation of bias and variability for the BB model. Concurrent over- and underestimation of discharges over the time series result in a good water balance: close to 1 for $\beta$ and $\overline{B_{rel}}$ and 0 for $\beta_n$. Depending on the criterion, the variability parameter can also affect the score in a similar counter-intuitive manner. $\alpha$ is heavily impacted by the counterbalance, whereas it seems mitigated for $\gamma, \alpha_{NP}$ and $|B_{area}|$. The timing parameters ($r$ and $r_s$) have an expected score that favour the BG model. However, the score difference on timing errors between BB and BG

models is very small (0.03 at best for $r$). The impact on the overall score is thus minimised compared to the one induced by bias and variability parameters, which can be cumulated (e.g. both $\beta$ and $\alpha$ counterbalancing errors in the KGE) or have a larger difference – up to 0.12 for $\alpha$. Counterbalancing errors can thus result in better values for bias and variability, which increase the overall score. In this case, the highest score may not be the most appropriate indicator of model relevance.

The largest differences in score appear for the LME and the LCE criteria as all their parameters are affected by

counterbalancing errors ($\beta, r\alpha$ and $r/\alpha$). The KGE and the KGE'' also show significant differences as they accumulate the counterbalancing errors of $\alpha$ and $\beta$. The KGE' demonstrates a smaller difference than the KGE due to the use of $\gamma$. Both FDC-based criteria KGE$_{NP}$ and DE' show the smallest differences due to $\alpha_{NP}$ and $|B_{area}|$, which have a nearly equal value for both BB and BG models. The NSE has a slightly better score on the BG model, while the difference is more pronounced on d$_r$.

This example demonstrates how relative error metrics can cancel out each other and affect the design and the evaluation of hydrological models. The counterbalancing errors especially affect bias parameters ($\beta, \beta_n$ and $\overline{B_{rel}}$) but also the variability parameter $\alpha$.

### 3.3    Exploring counterbalancing errors with synthetic transformations

Figure 4 shows the score distribution of the synthetic set of hydrographs presented in Sect. 3.1. For each value of $\omega_1$, the

minimum and maximum criteria scores of the transformations resulting from all combinations with $\omega_2$ provide the dashed envelope of the score distribution, with the maximum transformation score at the top (1 corresponding to a perfect model), and the worst at the bottom. The transformations corresponding to the BG models (with $\omega_2 = 1$) are represented by the black line. All transformations included in the dashed envelope can be identified as "Bad-Bad" models, except when $\omega_1 = 1$ or $\omega_2 = 1$ (black line).








**Figure 4: Score of each transformation for all $[\omega_1; \omega_2]$ combinations by performance criteria.**

It is obvious that the KGE and its variants – KGE', KGE'', $KGE_{NP}$, DE', LME and LCE – always evaluate one or several BB models as better than the BG model for a same $\omega_1$ value, except for $\omega_1 = 1$. On the other hand, the NSE and the $d_r$
correctly identify the BG model as the best transformation for all combinations of $[\omega_1; \omega_2]$, i.e. the black line is always above the dashed envelope. The envelope of the KGE, KGE' and KGE'' criteria are similar, but they do not display the same difference between the best scores and the scores of the BG models. These differences are smaller for the latter two because the KGE' is based on $\gamma$ instead of $\alpha$, and the KGE'' is based on $\beta_n$ instead of $\beta$, for which it is demonstrated in Sect. 3.2 that they both soften counterbalancing errors. The envelope of the LCE criterion looks like that of the KGE. However, the
difference between the best scores and the scores of the BG models is much higher. This is likely due to the nature of the





equation consisting in 3 parameters affected by counterbalancing errors ($\beta$, $r\alpha$ and $r/\alpha$). The LME criterion has a very distinctive envelope, for which the maximum score of 1 is reached for a lot of BB models, even when both $\omega_1$ and $\omega_2$ are different from 1. This can be explained by the interaction between $r$ and $\alpha$ that leads to an infinite number of solutions (Choi, 2022). The KGE$_{NP}$ and the DE' (FDC-based criteria) both shows similar envelopes with a break point near the

maximum transformation score in both ways around $\omega_1 = 1$. This is especially pronounced for the DE', for which the BG model is nearly the best model between $\omega_1 = 0.83$ and $\omega_1 = 1.17$. These results show that counterbalancing errors can happen on a large range of parameters, and when using the KGE or its variants, there is a possibility for the more meaningful model (i.e. BG model) to have a lower score than a "compensated" or "Bad-Bad" model.

<figcaption>Figure 5: Graph of each $[\omega_1; \omega_2]$ combination identified as the best transformation by each performance criteria. The NSE and the d$_r$ black lines coincide at $\omega_2 = 1$.</figcaption>


Figure 5 shows the value of $\omega_2$ corresponding to the best evaluation for a given $\omega_1$, by performance criteria. As identified above, the NSE and the $d_r$ both evaluate the BG models as the best transformations (NSE and $d_r$ black lines coincide at $\omega_2 =$ 1, Fig. 5). Counterbalancing errors are apparent for the KGE and its variants. For $\omega_1 \neq 1$, best transformations are always BB models and follow two conditions: (i) if $\omega_1 < 1$ then $\omega_2 > 1$, and (ii) if $\omega_1 > 1$ then $\omega_2 < 1$. This means that, in this case, such performance criteria will always be flawed towards concurrent under- and overestimation of discharges in a transformation.

## 4    Real case study

To highlight how counterbalancing errors can affect the assessment of hydrological models on a real case study, we used two different modelling approaches: artificial neural networks (ANN) and reservoir models. The simulations of karst spring discharges of both models were evaluated on the same 1-year validation period. To clearly highlight the problem, we deliberately chose a reservoir simulation that is noticeably affected by counterbalancing errors – yet still realistic. Further information on the modelling approaches, the input data, the calibration strategy and the simulation procedure can be found in Cinkus et al. (2022).

### 4.1    Study site

The Unica springs are the outlet of a complex karstic system influenced by a network of poljes. The recharge area is about 820 km² and is located in a moderate continental climate with a strong snow influence. Recharge comes from both (i) allogenic infiltration from two sub-basins drained by sinking rivers, and (ii) autogenic infiltration through a highly karstified limestone plateau (Gabrovšek et al., 2010; Kovačič, 2010; Petric, 2010). The network of connected poljes constitutes a common hydrological entity that induces a high hydrological variability in the system, and long and delayed high discharges at the Unica springs (Mayaud et al., 2019). The limestone massif can reach a height of 1800 m above sea level and has significant groundwater resources (Ravbar et al., 2012). A polje downstream of the springs can flood when the Unica discharge exceeds 60 m³ s⁻¹ for several days. If the flow reaches 80 m³ s⁻¹, the flooding can reach the gauging station and influence its measurement. The flow data are from the gauging station in Unica-Hasberg (ARSO, 2021a). Precipitation, height of snow cover, and height of new snow data are from the meteorological stations in Postojna and Cerknica (ARSO, 2021b). Temperature and relative humidity data are from the Postojna station. Potential evapotranspiration is calculated from the Postojna station data with the Penman-Monteith formula (Allen et al., 1998).

### 4.2    Modelling approaches

The first modelling approach is based on Convolutional Neural Networks (CNN) (LeCun et al., 2015), which is a specific type of ANN that is powerful in processing image-like data but also very useful for processing sequential data. The model consists of a single 1D Convolutional layer with a fixed kernel size of three and an optimised number of filters. This layer





was complemented by a Max-Pooling layer a Monte-Carlo dropout layer with 10% dropout rate and two dense layers. The first dense layer has an optimised number of neurons and the second a single output neuron. We programmed our models in
Python 3.8 (van Rossum, 1995), using the following frameworks and libraries: BayesOpt (Nogueira, 2014), Matplotlib (Hunter, 2007), Numpy (van der Walt et al., 2011), Pandas (Reback et al., 2021; McKinney, 2010), Scikit-Learn (Pedregosa et al., 2018), TensorFlow 2.7 (Abadi et al., 2016) and its Keras API (Chollet et al., 2015).

The second modelling approach is a reservoir model, which is a conceptual representation of a hydrosystem consisting of several reservoirs that are supposed to be representative of the main processes involved. We used the adjustable modelling
platform KarstMod (Mazzilli et al., 2019). The model structure consists of one upper reservoir for simulating soil and epikarst processes (including a soil available water capacity), and two lower reservoirs corresponding to matrix and conduits compartments. A very reactive transfer function from the upper reservoir to the spring is used to reproduce very fast flows occurring in the system.

### 4.3 Impact of counterbalancing errors on model evaluation

Figure 6a shows the results of the two hydrological models on Unica springs. The models have overall good dynamics and succeed to reproduce the observed discharges. Regarding high flow periods, both models show a small timing error, inducing a delay in the simulated peak flood. The two first flood events (February-March 2017) are slightly underestimated by the ANN model while the first peak is overestimated by the reservoir model. Although having a similar volume estimate, the third flood event (May 2017) is better simulated by the ANN model because (i) the timing error is less important and (ii) the
recession period is accurate. The last flood event (September 2017) comprise a small peak followed by a very high and long-lasting flood. Both models fail to account for the small peak. The following important flood event is highly overestimated by the reservoir model, while being nicely simulated by the ANN model – despite the small underestimation and timing error. The small flood events (mid-January, mid-April, early and late June 2017) are better simulated by the ANN model than the reservoir model. The ANN model simulates them satisfactorily, except for the second one, where the simulated discharges
are overestimated. The reservoir model does not simulate the first two events at all and largely overestimates the last two, in addition to timing errors. Both models can be improved during recession and low flow periods. The ANN model is rather close to the observed discharges but seems to be too sensitive to precipitation (continuous oscillations). On the other hand, the reservoir model shows no oscillations but either overestimates or underestimates the observed discharges. In general, the ANN model can be described as better because it is closer to the observed values in the high and low flow periods. Some
events are not well simulated by both models (e.g. the May 2017 flood), which may be due to uncertainties in the input data.





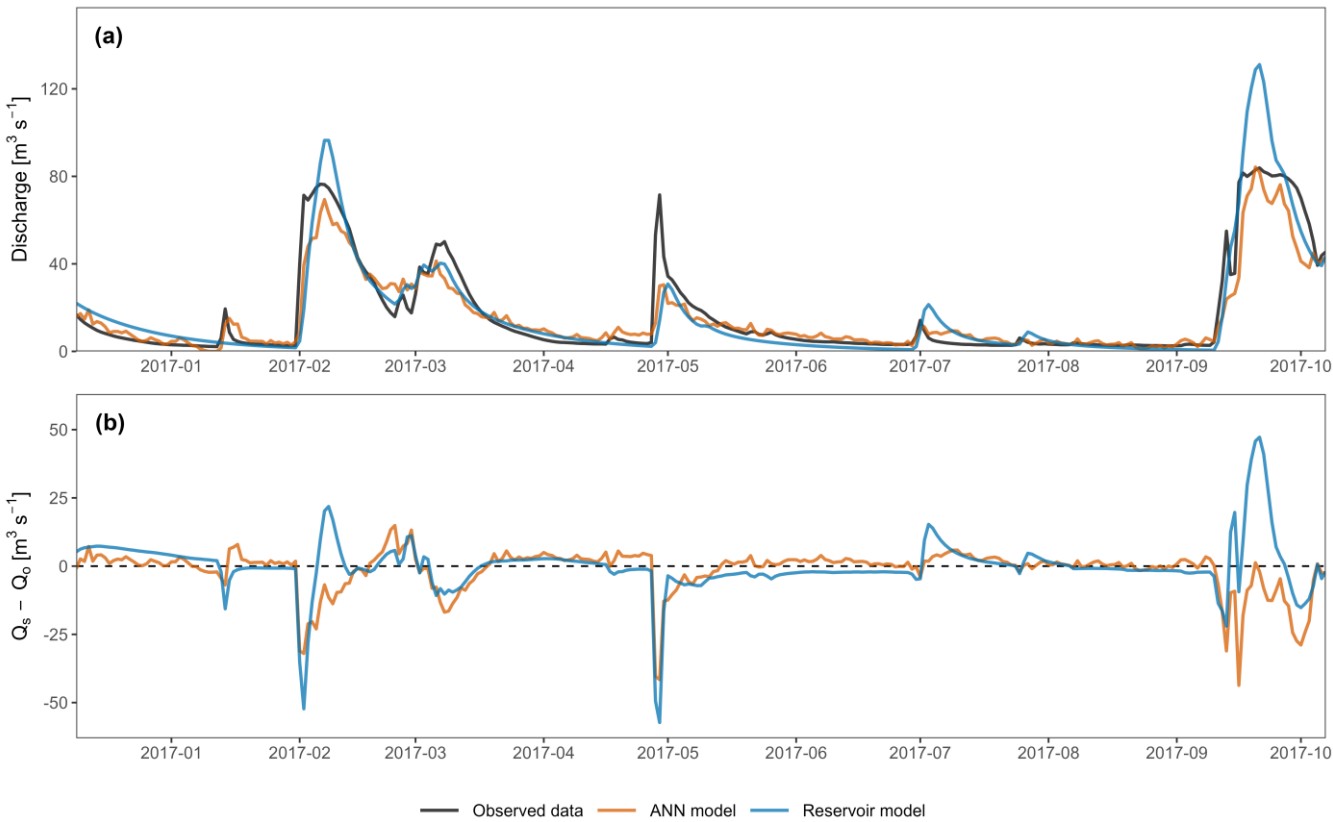

**Figure 6: (a) Observed and simulated spring discharge time series on the validation period. (b) Relative difference between simulated and observed discharge on the validation period.**

This visual assessment is confirmed only by few performance criteria: the NSE, $d_r$ and $KGE_{NP}$. These criteria evaluate the ANN model as better, although the performances of both models are quite close for the $d_r$. However, the KGE and most its variants (except the $KGE_{NP}$) all favour the reservoir model over the ANN model – sometimes by a large margin. It is interesting to note how similar these results are to those of the synthetic example (Fig. 3a). Looking at the values of the equations' parameters, we find that bias parameters are systematically better for the reservoir model, with 1 over 0.92 for $\beta$,

0 over -0.06 for $\beta_n$ and -0.07 over 0.18 for $\overline{B_{rel}}$. Timing errors are systematically better for the ANN model, with 0.95 over 0.92 for $r$ and 0.94 over 0.83 for $r_s$. Variability parameters favour the reservoir model with 1.1 over 0.78 for $\alpha$, 1.1 over 0.85 for $\gamma$, 0.22 over 0.3 for $|B_{area}|$, and a very close better value by 0.005 on the $\alpha_{NP}$ parameter. In summary, all bias and variability parameters have better values for the reservoir model, while timing and shape parameters are better for the ANN model.





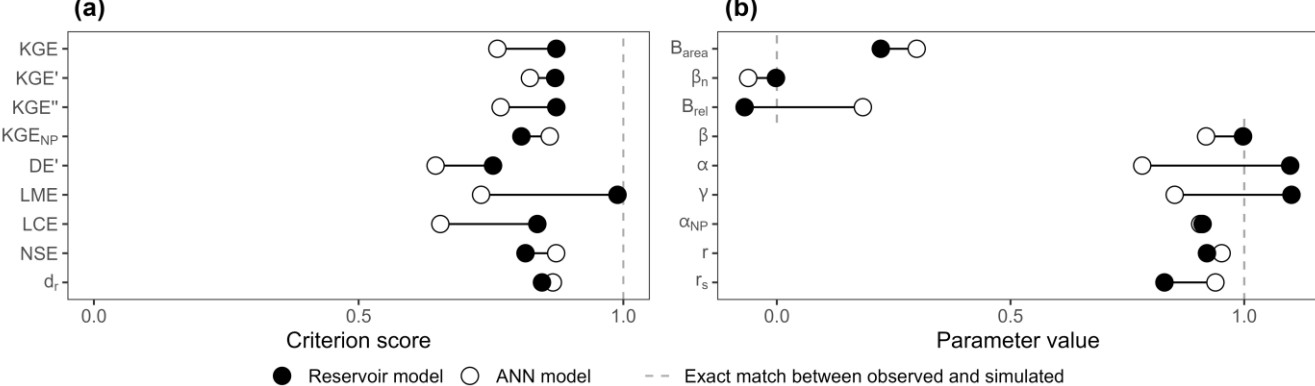


**Figure 7: (a) Score of the ANN and reservoir models according to the different performance criteria. (b) Values of the parameters used in the calculation of the performance criteria.**

As the KGE and its variants are generally composed of equally-weighted bias, variability and timing, their overall score is
heavily affected by compensation effects – except in the case of a large error on one parameter. In our case, all parameters have similar errors, which results in a better KGE for the reservoir model compared to the ANN model. This applies to all the KGE variants except the $KGE_{NP}$ where the error on $r_s$ is significant, resulting in a better score for the ANN model. The LME score is extremely high (0.99) for the reservoir model, which is probably due to the compensation of $r$ and $\alpha$ identified by Choi (2022). Also, using $\gamma$ instead of $\alpha$ for assessing the variability seems to lower counterbalancing errors.
Figure 6b shows that there is a consistent greater or equal overestimation of the reservoir model compared to the ANN model, except for the May-June period where the difference is small and insignificant compared to the February or September events. The underestimated values are similar for both approaches, except when the reservoir model overestimates the flooding events. Interestingly, the cumulative sum of the absolute bias error between simulated and observed values is smaller for the ANN model (1394 $m^3$) than the reservoir model (1611 $m^3$), but still the relative bias and
variability parameters are better for the reservoir model. This observation highlights how counterbalancing errors can impair the evaluation of hydrological models: seemingly better parameters values (bias and variability) that increase criteria scores are not necessarily associated with an increase in model relevance.

## 5    Recommendations

The aim of this paper is primarily to raise awareness among modellers. Performance criteria generally comprise several
aspects of the characteristics of a model into a single value, which can lead to an inaccurate assessment of said aspects. Ultimately, all criteria have their flaws and should be carefully selected with regards to the aim of the model.



## 5.1 Use of relevant performance criteria

Table 1 summarises the presence and impact of counterbalancing errors, as well as the advantages and drawbacks (as reported in other studies) of the different performance criteria. The recommendations on counterbalancing errors are based on the results of this research – i.e. synthetic and real case studies. The KGE and all its variants are affected by counterbalancing errors with varying degrees of intensity: (i) mildly impacted (+) for the KGE', $KGE_{NP}$ and DE, (ii) moderately impacted (++) for the KGE, KGE'' and LCE, and (iii) strongly impacted (+++) for the LME. In this study, the NSE and $d_r$ stand out as clearly better since they have no counterbalancing errors. However, they have other drawbacks that are not associated with counterbalancing errors but still important to consider. We thus recommend using performance criteria that are not or less prone to counterbalancing errors (NSE, $d_r$, KGE', $KGE_{NP}$, DE), preferably in a multi-criteria framework to better quantify the different aspects of a hydrological model and further reduce the uncertainties inherent to each performance criterion.

**Table 1: Presence and impact of counterbalancing errors (CE) on the assessment of model performance of different performance criteria. The impact of CE is denoted as null (/), mild (+), moderate (++), or strong (+++).**



| Criterion | Year | Affected by CE | Impact of CE | Advantages | Drawbacks[a] |
|---|---|---|---|---|---|
| KGE | 2009 | Yes | ++ | Variability is not underestimated (Gupta et al., 2009) | Still slight underestimation of high discharges (Gupta et al., 2009) |
| | | | | | Bias and variability are cross correlated (Kling et al., 2012) |
| | | | | | Implicit assumptions of data linearity, data normality and absence of outliers (Pool et al., 2018) |
| | | | | | No inherent benchmark (Knoben et al., 2019) |
| | | | | | Not suited to logarithmic transformation of discharge (Santos et al., 2018) |
| KGE' | 2012 | Yes | + | Bias and variability are not cross correlated (Kling et al., 2012) | |
| KGE'' | 2021 | Yes | ++ | The score is not overly sensitive to mean values close to zero (Santos et al., 2018; Tang et al., 2021) | |
| $KGE_{NP}$ | 2018 | Yes | + | Reduce the impact of implicit assumptions of data linearity, data normality and absence of outliers by using non-parametric parameters (Pool et al., 2018) | |
| DE | 2021 | Yes | + | Aims to provide a stronger link to hydrological processes (Schwemmle et al., 2021) | |
| LME | 2020 | Yes | +++ | Improve the simulation of extreme events (Liu, 2020) | Infinite number of solutions for the maximum score (Lee and Choi, 2022) |
| | | | | | Inclination to overestimate high flows and underestimates low flows (Lee and Choi, 2022) |
| LCE | 2022 | Yes | ++ | Improve the simulation of extreme events (Lee and Choi, 2022) | |
| NSE | 1970 | No | / | | The contribution of $\beta_n$ depends on the variability (Gupta et al., 2009) |
| | | | | | Variability is underestimated (Gupta et al., 2009) |
| | | | | | The benchmark is inappropriate for highly variable discharges (Gupta et al., 2009) |
| $d_r$ | 2012 | No | / | Address the shortcomings of the NSE (Jackson et al., 2019, Willmott et al., 2012) | |

[a]KGE drawbacks may likely apply to KGE variants, but this hasn't been studied extensively





## 5.2 Use of scaling factors

The assessment of the hydrological models in the real case study shows how concurrent over- and underestimation can generate counterbalancing errors on bias and variability parameters. For the case study considered in this paper, the ANN model, although offering a better simulation, is evaluated as – sometimes considerably – worse than the reservoir model, because it slightly underestimates the total volume. This has a great impact on the overall score, as the KGE and its variant are calculated with both bias and variability parameters accounting for 2/3 of the overall criterion score.

While the overall balance (bias) may be a desired feature in a model, we showed that a good value may be accidental and result from counterbalancing errors. The common use of the KGE neglects one of the original proposals which is to weight the parameters $\beta$, $\alpha$ and $r$ in the equation. Gupta et al. (2009) proposed an alternative equation for adjusting the emphasis on the different aspects of a model:

$$KGE_s = 1 - \sqrt{[s_\alpha(\alpha - 1)]^2 + \left[s_\beta(\beta - 1)\right]^2 + [s_r(r - 1)]^2} \tag{22}$$

with $s_r$, $s_\beta$ and $s_\alpha$ the scaling factors of $r$, $\beta$ and $\alpha$, respectively. By default, these factors are equal to 1, which induces a weight of 1/3 on the parameter in absolute value ($r$) and 2/3 on the parameters in relative values ($\beta$, $\alpha$). To the best of our knowledge, only Mizukami et al. (2019) ever considered changing the scaling factors when using the KGE. We suggest to carefully consider such scaling factors for the calibration and the evaluation of hydrological models using the KGE and its variants. Depending on the purpose of the model, they can help to emphasise particular aspects of a model or reduce the influence of relative parameters and counterbalancing errors.

Figure 8 shows how emphasising absolute parameters with scaling factors helps to reduce the influence of counterbalancing errors for the KGE (Fig. 8a) and its most used variant KGE' (Fig. 8b). The default value (1-1-1) – corresponding to scaling factors of 1 for $r$, 1 for $\alpha$ (KGE) or $\gamma$ (KGE') and 1 for $\beta$, respectively – is compared to other factor combinations with different ratios between absolute and relative parameters. The 1:2 ratio (1-2-2) increases counterbalancing errors as the emphasis is on the relative parameters, while the 2:1, 3:1, 4:1, and 5:1 ratios decrease counterbalancing errors. The ANN model is evaluated as better with the 4:1 ratio for the KGE and the 3:1 ratio for the KGE', highlighting that the KGE' is less sensitive to counterbalancing errors. This also shows how the score of a performance criterion and by extension its interpretation can be radically different depending on the parameters used in the equation. This is why a multi-criteria framework can strengthen the evaluation of models and reduce the uncertainties of performance criteria scores.





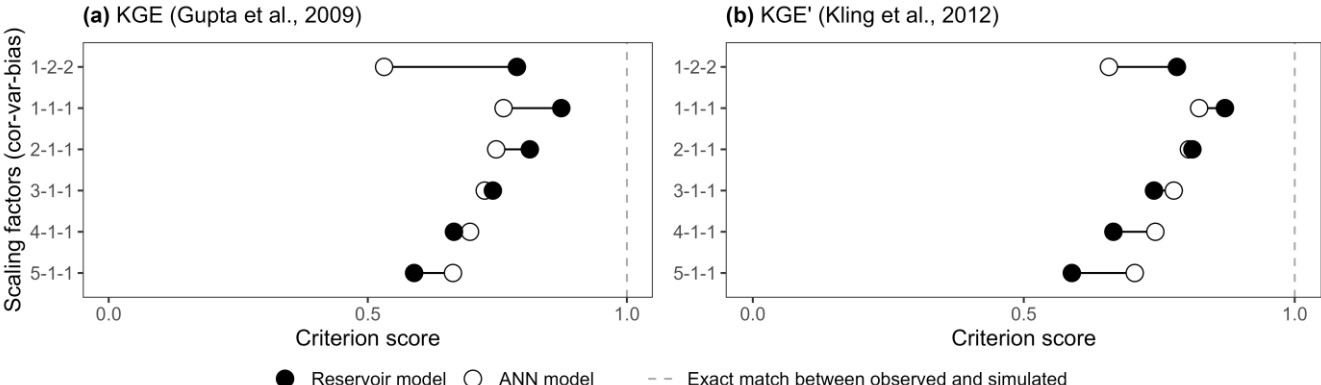

**Figure 8: (a) KGE and (b) KGE' scores of the ANN and reservoir models (Fig. 6a) according to different scaling factors. The y-axis numbers correspond to the scaling factors of the timing, variability and bias parameters, with the default being 1-1-1.**

## 6    Conclusion

This study sets out to explore the influence of counterbalancing errors and raise awareness among modellers about the use of performance criteria for calibrating and evaluating hydrological models. A total of nine performance criteria (NSE, KGE, KGE', KGE'', $KGE_{NP}$, DE, LME, LCE and $d_r$) are analysed. The investigation of synthetic time series and real hydrological models shows that concurrent over- and underestimation of multiple parts of a discharge time series may favour bias and variability parameters. This especially concerns the bias parameters ($\beta$, $\beta_n$ and $B_{rel}$) as their values are all influenced by

counterbalancing errors in both synthetic time series and the real case study. On the other hand, the impact of counterbalancing errors on the variability parameters seems to depend on the time series: only the value of $\alpha$ is influenced in the synthetic time series, while the values of all variability parameters ($\alpha, \gamma, |B_{area}|$ and $\alpha_{NP}$) are influenced in the real hydrological models. As bias and variability parameters generally account for 2/3 of the weight in the equation of certain performance criteria, this can lead to an overall higher criterion score without being associated to an increase in model

relevance. This is especially concerning for the KGE and its variants, as they generally use relative parameters for evaluating bias and variability in hydrological models. These findings highlight the importance of carefully choosing a performance criterion adapted to the purpose of the model. Recommendations also include to use scaling factors to emphasise different aspects of a hydrological model and reduce the influence of relative parameters on the overall score of the performance criterion. Further research could explore the appropriate values of scaling factors to use, depending on the modelling

approach and the purpose of the study.

**Code and data availability**

We provide complete scripts for reproducing the results on the synthetic time series (Sect. 3), as well as ANN model code and KarstMod .*properties* file (reservoir model) on GitHub (Cinkus and Wunsch, 2022). Unica spring discharge time series and meteorological data are available from the Slovenian Environment Agency (ARSO, 2021a, b).

**Author contribution**

GC, NM and HJ conceptualised the study and designed the methodology. GC and AW developed the software code. GC performed the experiments and investigated and visualised the results. AW provided the ANN results for the case study. GC wrote the original paper draft with contributions from AW and NR. All the authors contributed to the interpretation of the results and review and editing of the paper draft. NM and HJ supervised the work.

**Competing interests**

The authors declare that they have no conflict of interest.

**Acknowledgments**

We thank the French Ministry of Higher Education and Research for the thesis scholarship of G. Cinkus as well as the European Commission for its support through the Partnership for Research and Innovation in the Mediterranean Area
(PRIMA) program under Horizon 2020 (KARMA project, grant agreement number 01DH19022A). We further thank the Slovenian Research Agency for financial support within the project Infiltration processes in forested karst aquifers under changing environment (No. J2-1743). For the data provided, we also acknowledge the Slovenian Environment Agency (ARSO, 2021a, b).

The analyses were performed using R (R Core Team, 2021) and the following packages: readxl, readr, dplyr, tidyr, ggplot2,
lubridate (Wickham et al., 2019), cowplot (Wilke, 2020), diag-eff (Schwemmle et al., 2021), flextable (Gohel, 2021), hydroGOF (Mauricio Zambrano-Bigiarini, 2020), HydroErr (Roberts et al., 2018) and padr (Thoen, 2021). The manuscript was written with the Rmarkdown framework (Allaire et al., 2021; Xie et al., 2018, 2020).



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
