# Peer review of "When best is the enemy of good – critical evaluation of performance criteria in hydrological models"

_Hydrology and Earth System Sciences, 2022_

## Community Comment (CC2)

**The objective function, Nash–Sutcliffe efficiency scale, and autoregressive projections**

Parameter optimization of a hydrologic model needs to specify an objective or penalty function for the model to meet.

The classical Nash and Sutcliffe 1970 efficiency scale (NSE) expressed by Eq. (10) can be recast using the original notation as: $R^2 = 1 - F/F_0$. It has both an objective function in residual variance $F$, which is sum of squares of the simulation error (SSE) and an observed-mean-flow ($\mu_o$) benchmark embedded in initial variance $F_0$, a fixed value. There is a one-to-one correspondence between NSE and $F$, and optimizing NSE is same as optimizing $F$. But this is not necessarily true in its variants, including an earliest known one, Ding 1974, Eqs. (40) and (47) therein.

NSE is a measure of correlation as well as others between simulation and observation as shown in a componentized form in Eq. (11). What it needs physically as well as statistically is at least one auxiliary benchmark to help interpret its intermediate scores between a perfect score of 1 for an observed or reference hydrograph, i.e. a perfect model, and of 0 for the (primary) benchmark model, $\mu_o$. Establishing auxiliary benchmarks or baselines will help address one question about the popular performance metric: how close to 1 are NSE values reachable by models, e.g., Nearing et al. 2022, Table 1 therein.

The concept of two-parameter ($\omega 1{:}\omega 2$) homothetic transformation hydrographs represents a first step toward searching for such auxiliary benchmarks, as described for a twin-peak synthetic hydrograph in Sections 3.1, 3.2, Equation (21), and presented in Figures 1, 2 and 3.

I've put forward a simplest second-order autogressive process of the streamflow, $AR(2, c = 0, c_1 = 2, c_2 = -1)$, as a replacement of the primary benchmark, $\mu_0$, e.g., Ding 2018. This, a slope-based projection hydrograph, instead could be considered a secondary benchmark, e.g., Azmi et al. 2021, SC1 and AC1 therein. In the same vein, a simplest third-order $AR(3, 0, 2, -2, 1)$, a curvature-based projection hydrograph, could be a tertiary one.

AR(2) and AR(3) projection hydrographs can be generated for the twin-peak example hydrograph. Scoring them would yield NSE values, calibration free.

I encourage the authors to pursue this AR projection approach in a future study. For the example hydrograph, I for one would be interested in what are NSE scores for AR(2) and AR(3) benchmarks, and whether the higher score of the two is lower than but close to the values shown in Fig. 3(a) for both BB (Bad-Bad) and BG(Bad-Good) transformations.

**References**

Azmi, E. et al. (2021). "Technical note: "Bit by bit": a practical and general approach for evaluating model computational complexity vs. model performance". In: *Hydrology and Earth System Sciences* 25.2, pp. 1103–1115. DOI: 10.5194/hess-25-1103-2021. URL: https://hess.copernicus.org/articles/25/1103/2021/.

Ding, J. (2018). "Interactive comment on "On the choice of calibration metrics for "high flow" estimation using hydrologic models" by Naoki Mizukami et al." In: DOI: https://doi.org/10.5194/hess-2018-391-SC1.

Ding, JY (1974). "Variable unit hydrograph". In: *Journal of Hydrology* 22.1-2, pp. 53–69.

Nash, J Eamonn and Jonh V Sutcliffe (1970). "River flow forecasting through conceptual models part I—A discussion of principles". In: *Journal of hydrology* 10.3, pp. 282–290.

Nearing, G. S. et al. (2022). "Technical note: Data assimilation and autoregression for using near-real-time streamflow observations in long short-term memory networks". In: *Hydrology and Earth System Sciences* 26.21, pp. 5493–5513. DOI: 10.5194/hess-26-5493-2022. URL: https://hess.copernicus.org/articles/26/5493/2022/.

---

## Author Comment (AC1)

**hess-2022-380: CC1**

**The study investigated counterbalancing errors which are inherent in Nash-Sutcliffe Efficiency and its variants. Reliability of the performance criteria is important to boost confidence with which a particular model can be chosen. There are some important points which the authors could address to strengthen their paper.**

We would like to thank you for taking the time to carefully review our manuscript and for providing valuable comments and suggestions.

1. **In the last sentence of the abstract, the authors mention the use of multi-criteria framework in their recommendation. On the need to consider a particular "goodness-of-fit" metric within the multi-criteria framework, the authors could clarify on other specific requirements apart from the general condition that the performance criteria should be less or not prone to counterbalancing.**

   **Furthermore, the use of several criteria for a particular calibration can complicate the applications of automation of famous search strategies or algorithms (Onyutha 2022). It is upon this basis that a number of performance criteria which are not mathematically and statistically related tend to be formed into single metric. For instance, Kling-Gupta Efficience combines three components including measures of bias, variability and linear correlation between observed (X) and modelled (Y) series. Thus, the authors should provide more considered justification for their recommendation of the use of multi-criteria framework for calibration of hydrological models.**

Indeed, the explanation around the use of a multi-criteria framework is unclear and we will clarify this aspect. As you mentioned, the KGE already includes several components for evaluating the bias, variability and linear correlation of a model, which help to evaluate different aspects of a model. We are currently considering providing additional guidelines for evaluating "how much" a model is prone to counterbalancing errors, which can help to assess the relevance of the performance criteria used for the calibration and evaluation.

2. **Most of (if not all) the metrics used in this study rely on the assumption that X and Y are linearly related. Note that X and Y can be so highly**

   **dependent yet it may be nearly impossible to detect the dependence using classical dependence metric (Székely et al. 2007). In other words, the authors should clarify on whether the model performance results of this study may not have been affected by the said assumption.**

Thank you for pointing out this implicit assumption of the KGE and its variants. Although this study focuses on counterbalancing errors in widely used performance criteria and not (so much) on the correlation between X and Y, it is important to clarify whether the data linearity between X and Y is skewed – which is often the case in hydrological modelling – to better appreciate the model performance results. We will add some text (and maybe some scatter plots) on this aspect. Note that we also included the non-parametric KGE (Pool et al., 2018), which is based on the Spearman correlation coefficient and the flow duration curve, and has no assumption of data linearity.

Pool, S., Vis, M., and Seibert, J.: Evaluating model performance: towards a non-parametric variant of the Kling-Gupta efficiency, Hydrol. Sci. J., 63, 1941–1953, https://doi.org/10.1080/02626667.2018.1552002, 2018.

3. **Most of the performance criteria (especially Nash Sutcliffe Efficiency NSE (Nash and Sutcliffe, 1970) and its variants) comprise some forms of the well-known coefficient of determination (R-squared) (see Onyutha, 2022). R-squared is known to have various short comings. To address these short comings, new metrics including the revised R-squared (RRS) and hydrological model skill score E (Onyutha 2022) were developed. Thus, instead of focussing on NSE and its variants, the authors should compare results of many other performance criteria such as RRS and E. Accordingly, Figure 7 and Table 1 in this manuscript can be updated. The MATLAB codes to compute RRS and E can be downloaded via https://doi.org/10.5281/zenodo.6570905 and the codes can also be found as supplementary material to the paper by Onyutha (2022).**

Thank you for the suggestion of these two innovative performance criteria and the associated code. We will really consider adding them in the study as they aim to address the shortcomings of widely used performance criteria. It can be interesting to see if these metrics are prone to counterbalancing errors.

4. **ON EQUATION 20**

   **According to Legates & McCabe (2013), the refinement of Index of Agreement (IOA) (Willmott, 1981) made by Willmott et al. (2012) especially regarding the extension of the IOA bound from 1 to 0 was unnecessary. Check Legates & McCabe (2013) for other limitations of the refined IOA. Therefore, could the authors make use of the original form of IOA for their model performance evaluation and analyses?**

Thank you for pointing out this interesting discussion about the refined index of agreement. We will make use of the original for of the index of agreement in the revised version of the manuscript.

---

## Author Comment (AC2)

**hess-2022-380: CC2**

**Parameter optimization of a hydrologic model needs to specify an objective or penalty function for the model to meet.**

**The classical Nash and Sutcliffe 1970 efficiency scale (NSE) expressed by Eq. (10) can be recast using the original notation as: $R^2 = 1 - F/F_0$. It has both an objective function in residual variance $F$, which is sum of squares of the simulation error (SSE) and an observed-mean-flow ($\mu_0$) benchmark embedded in initial variance $F_0$, a fixed value. There is a one-to-one correspondence between NSE and $F$, and optimizing NSE is same as optimizing $F$. But this is not necessarily true in its variants, including an earliest known one, Ding 1974, Eqs. (40) and (47) therein.**

**NSE is a measure of correlation as well as others between simulation and observation as shown in a componentized form in Eq. (11). What it needs physically as well as statistically is at least one auxiliary benchmark to help interpret its intermediate scores between a perfect score of 1 for an observed or reference hydrograph, i.e. a perfect model, and of 0 for the (primary) benchmark model, $\mu_0$ Establishing auxiliary benchmarks or baselines will help address one question about the popular performance metric: how close to 1 are NSE values reachable by models, e.g., Nearing et al. 2022, Table 1 therein.**

**The concept of two-parameter ($\omega_1$:$\omega2$) homothetic transformation hydrographs represents a first step toward searching for such auxiliary benchmarks, as described for a twin-peak synthetic hydrograph in Sections 3.1, 3.2, Equation (21), and presented in Figures 1, 2 and 3.**

**I've put forward a simplest second-order autogressive process of the streamflow, AR(2, c = 0, c1 = 2, c2 = −1), as a replacement of the primary benchmark, $\mu_0$, e.g., Ding 2018. This, a slope-based projection hydrograph, instead could be considered a secondary benchmark, e.g., Azmi et al. 2021, SC1 and AC1 therein. In the same vein, a simplest third-order AR(3, 0, 2, −2, 1), a curvature-based projection hydrograph, could be a tertiary one.**

**AR(2) and AR(3) projection hydrographs can be generated for the twin-peak example hydrograph. Scoring them would yield NSE values, calibration free.**

**I encourage the authors to pursue this AR projection approach in a future study. For the example hydrograph, I for one would be interested in what are NSE scores for AR(2) and AR(3) benchmarks, and whether the higher score of the two is lower than but close to the values shown in Fig. 3(a) for both BB (Bad-Bad) and BG(Bad-Good) transformations.**

Thank you for taking the time to review our manuscript and providing interesting explanations and suggestions about the use of autoregressive projections as a benchmark model.

We also read your comments in the peer-review section of Mizukami et al. (2019), Knoben et al. (2019), and Azmi et al. (2021), which were really insightful into the autoregressive projections. In a future study, it could be interesting to look for this aspect of performance criteria.

As you are interested in the results of the NSE scores for AR(2) and AR(3) benchmarks, we performed the calculation on the twin-peak example hydrograph, using the equations below:

$$Q_{AR2}(t) = 2 * Q_{obs}(t-1) - Q_{obs}(t-2)$$

$$Q_{AR3}(t) = 2 * Q_{obs}(t-1) - 2 * Q_{obs}(t-2) + Q_{obs}(t-3)$$

$$NSE_{AR} = 1 - \frac{\sum(x_s(t) - x_o(t))^2}{\sum(x_o(t) - Q_{AR}(t))^2}$$

The following graph shows the observed time series alongside the time series of the different benchmarks used, i.e. $AR(2)$ and $AR(3)$:

[Figure]

Note that, the synthetic time series of the example hydrograph has no value before $t = 0$, therefore the first three values of the autoregressive projections will be non-defined as $Q_{obs}(t-1)$, $Q_{obs}(t-2)$ and $Q_{obs}(t-3)$ are $NA$. Because of this, we evaluated the models on the whole synthetic time series except the first three values, as can be appreciated on the graph below:

[Figure]

The following graph shows the score of each *NSE*, $NSE_{AR2}$ and $NSE_{AR3}$ performance criteria on the BB and BG synthetic models:

[Figure]

The values of the NSE, NSE_AR2 and NSE_AR3 are detailed in the table below:

| Criterion | Bad-Bad model | Bad-Good model |
|---|---|---|
| NSE | 0.922 | 0.953 |
| NSE_AR2 | 0.866 | 0.918 |
| NSE_AR3 | 0.880 | 0.927 |

We can see that the NSE with AR(3) benchmark has a higher score than with AR(2), with 0.880 and 0.927 for the BB and BG models, respectively. The NSE evaluations with AR benchmarks still yield good scores, close to the NSE with $\mu_0$ benchmark. The score difference between the BB and BG models is slightly, but not significantly, higher for the NSE scores with AR benchmarks.

**References**

Azmi, E., Ehret, U., Weijs, S.V., Ruddell, B.L., Perdigão, R.A.P., 2021. Technical note: "Bit by bit": A practical and general approach for evaluating model computational complexity vs. Model performance. Hydrology and Earth System Sciences 25, 1103–1115. https://doi.org/10.5194/hess-25-1103-2021

Knoben, W.J.M., Freer, J.E., Woods, R.A., 2019. Technical note: Inherent benchmark or not? Comparing Nash and Kling efficiency scores. Hydrol. Earth Syst. Sci. 23, 4323–4331. https://doi.org/10.5194/hess-23-4323-2019

Mizukami, N., Rakovec, O., Newman, A.J., Clark, M.P., Wood, A.W., Gupta, H.V., Kumar, R., 2019. On the choice of calibration metrics for "high-flow" estimation using hydrologic models. Hydrol. Earth Syst. Sci. 23, 2601–2614. https://doi.org/10.5194/hess-23-2601-2019

---

## Author Comment (AC3)

**Having carried out hydrological modelling for the past 30 years it is interesting to see how the use of different performance criteria has developed. The Nash-Sutcliffe Efficiency (NSE) criteria has been the main criteria for flooding issues (and very often a general criteria on the overall performance of a model) for a long time. It has a number of well documented drawbacks but has the advantage of the values being widely understood. Kling-Gupta Efficiency (KGE) and its variants have become more popular recently but my feeling is that it is less well understood and some of the issues associated with its use have not been fully explored.**

**This paper is a useful addition to the subject of different performance criteria as it clearly shows that in the KGE there can be counterbalancing errors (i.e sometimes an over estimation and sometimes an under estimation of discharge) which produce a higher value without there being an improvement in the model. Whereas these counterbalancing error do not occur for the NSE. The authors summarize the issue and their contribution very well when they say "The aim of this paper is primarily to raise awareness among modellers. Performance criteria generally comprise several aspects of the characteristics of a model into a single value, which can lead to an inaccurate assessment of said aspects. Ultimately, all criteria have their flaws and should be carefully selected with regards to the aim of the model"**

**The paper is well written and presented. There is a good summary of the current state in the use of different performance criteria in hydrological models. The use of both a sythentic time series and a real case study gives more confidence in the issue of these counterbalancing errors. Overall, it is a good bit of work with a clear conclusion reached. I am happy to accept the paper with minor revisions**

Thank you very much for you for your positive feedback, and also for your careful review of the manuscript. We are pleased that you find the paper interesting, well-constructed and meaningful.

**Specific comments:**

- **I wonder if it would be useful to mention some of the benchmarking studies in hydrological modelling (e.g. Seibert et al. 2018) which I feel are a useful addition with regards to the performance criteria of the models.**

Thank you for pointing out this interesting publication. We agree that it is important to mention the possible use of more relevant benchmarks in hydrological modelling. We modified the manuscript accordingly.

L30: *"Improvements can be made by working on (i) input data, [...], (iv) model calibration (Beven, 2019), and also (v) appropriate benchmarks for assessing model performance (Seibert et al., 2018)."*

L45: *"In relation to the assessment of model performance, Seibert et al. (2018) argued that the current benchmarks poorly reflect what could and should be expected of a model. They suggested to define lower and upper benchmarks based on the performance of a simple bucket-type model with few parameters, using the same data set."*

Added reference: *"Seibert, J., Vis, M. J. P., Lewis, E., and van Meerveld, H. j.: Upper and lower benchmarks in hydrological modelling, Hydrological Processes, 32, 1120–1125, https://doi.org/10.1002/hyp.11476, 2018."*

- **L98 (Equation 11). This is the Gupta(2009) equation. This is surely wrong as the last term should be minus not plus.**

Indeed, you are right, thanks for pointing this out. We corrected it.

- **L133-135. I do not understand this bit. I can see there are 361 transformation between -0.36 and 0.36 but I need not understand where the logarithmic scale comes in and how you get from here to the w values**

We wanted to study counterbalancing errors on a set of homothetic transformations ranging from roughly half to twice the discharges of the reference time series, which correspond to ω values of about 0.5 to 2. A linear sampling would have been uneven between underestimated and overestimated transformations (0.5–1 have a lower sample rate than 1–2). Sampling on the log-transformed interval allows to have an even distribution of the values below and above ω=1. As demonstrated in Eq. (1) and Eq. (2), the common logarithmic transformations (base 10) of 0.5 and 2 nearly equal -0.30103 and 0.30103, respectively; these values are equidistant to 0, which is the common logarithmic transformation of 1 (Eq. (3)).

$$\log_{10}(0.5) \approx -0.30103 \tag{1}$$
$$\log_{10}(2) \approx 0.30103 \tag{2}$$
$$\log_{10}(1) = 0 \tag{3}$$

We decided to take a slightly larger interval to ease the reading of the graphs, i.e. [-0.36, 0.36], sampled at a 0.002 step. The exponentiation in base 10 of the sampled values allows to get an even distribution of ω values around ω=1:

$$10^{-0.36} \approx 0.4365158 \tag{4}$$
$$10^{0.36} \approx 2.290868 \tag{5}$$
$$10^{0} = 1 \tag{6}$$

As $\frac{0.36}{0.002} = 180$, there are 361 transformations in total:

- 180 transformations below the ω=1 homothety in the [-0.36; -0.002] interval, with the minimum ω for -0.36 (Eq. (4))
- 180 transformations above the ω=1 homothety in the [0.002; 0.36] interval, with the maximum ω for 0.36 (Eq. (5))
- 1 transformation corresponding to the ω=1 homothety (Eq. (6))

We modified the manuscript with a better explanation of the logarithmic sampling procedure.

L133: *"ω values were sampled uniformly on the log-transformed interval [-0.36, 0.36] at a defined step of 0.002 to ensure a fair distribution between underestimated and overestimated transformations."*

L134: *"The exponentiation in base 10 of the sampled values results in 360 ω values evenly distributed around the $\omega = 1$ homothety, which corresponds to the reference time series (i.e. absence of transformation)."*

L136: *"We defined ω bounds such that the transformed peak discharge roughly ranges from half (ω ≈ 0.437 ≈ $10^{-0.36}$) to twice (ω ≈ 2.291 ≈ $10^{0.36}$) compared to the references time series."*

- **L195 (Figure 4). Would it be useful to also show the "Bad-Bad" model on this figure?**

Thank you, this is a great suggestion. It can be useful because it would show that, for some criteria, the "Bad-Bad" model is not even the most affected by counterbalancing errors.

[Figure]

- **Change "consisting in" to "consisting of"**
- **Maybe change "both ways" to "both sides" or "both directions"**
- **Change "succeed to reproduce" to "succeed in reproducing" or "successfully reproduce"**

Thank you for the corrections. We changed it in the revised manuscript.

- **L273-L274. "In general, the ANN model can be described as better because it is closer to the observed values in the high and low flow periods". As a hydrological modeller I agree the ANN model is better. But surely the whole point of performance criteria is to objectively decide which model is better. So how do you decide it is better when the performance criteria do not agree? There is no easy answer but I feel it is an important question that should be considered in more detail.**

You are questioning a valid point and we agree that it deserves a better and more detailed explanation. Thank you for considering the fact that the answer is not easy. We added a paragraph in the manuscript to better explain how the ANN model can be considered as better without using performance metrics (i.e. from a subjective assessment). We also improved the description of the models for the first and second flood events.

L262: *"The first flood event (February 2017) is slightly underestimated by the ANN model and highly overestimated by the bucket-type model. The second flood event (March 2017) is similarly underestimated by both models but the bucket-type model demonstrates a slightly better performance."*

L274: *"While this statement cannot be supported by performance metrics, we believe that an expert assessment based on intuition and experience is still valuable despite being intrinsically subjective. In this particular case, one can assess the main, distinctive flaws of each model: (i) the ANN model has continuous oscillations – especially on recession and low flow periods – and lacks of accuracy during recession periods; (ii) the bucket-type model highly overestimates several flood events and is inaccurate during a lot of recession and low flow periods. Figure 6b also shows that the ANN model has an overall lower bias than the bucket-type model. Hydrological models are generally used for (i) the prediction/forecast of water flood/inrush, (ii) the management of water resources, (iii) the characterisation of hydrosystems, and more recently (iv) the study of the impact of climate change on water resources. Most studies thus put the emphasis on extremes events (i.e. dry and flood periods), which in this case are more satisfactorily reproduced by the ANN model – in terms of volume estimate, timing and variability."*

- **There is no reference to Figure 7, should it be here.**

We added the reference to Figure 7 in the text.

L280: *"The visual assessment is confirmed only by a few performance criteria: the NSE, $d_r$ and $KGE_{NP}$ (Fig. 7a)"*

L283: *"Looking at the values of the equations' parameters (Fig. 7b), we find that [...]"*

- **L300-L303. I do not think this bit adds anything. I would remove these lines.**

You are right, this does not add a lot to the discussion, so we removed the sentences in the revised manuscript.

- **In Equation 22 the order of the parameters is alpha, beta, r. On Line 344 and subsequent lines it is r, alpha, beta. This is confusing. So when you look at (1-2-2) and you look at equation 22 everything needs to be swapped around as the 1 corresponds to r which is the last term in the equation**

Indeed, this is confusing. We changed the order of the y-axis number to be the same of the order of the equation, i.e. alpha, beta, r. The caption was edited: *"The y-axis numbers correspond to the scaling factors of the variability, bias and timing parameters, with the default being 1-1-1."*

[Figure]

- **Change "associated to" to "associated with"**
- **Change "include to" "include the"**

Thanks for the corrections. We changed it in the revised manuscript.

---

## Author Comment (AC4)

**The paper examines several goodness-of-fit, skill scores used for hydrologic model evaluations based on the synthetic flow data and real simulation data. In the paper, the types of the skill scores are classified into two—1) multi-variant-based skill scores such as KGE (the skill score computed based on multiple metrics like bias, variability error, correlation etc. using distance measures) and its variants and 2) NSE. The paper focused on the impacts on the skill scores (also its components - bias and variability error if applicable), originating from the situation where under- and over-estimation on the peak flow can exist in one hydrograph. Also, the paper discusses compensation between the components of the skill scores, namely bias and variability. The paper concludes KGE type scores can be inflated for the hydrograph include both under- and over-estimation of the event (because of lower bias and variability error over the time series), which does not necessarily represent "accurate" simulations. The paper suggests that weighting KGE components mitigates this misleading score values.**

**I think the hydrologic modeling community intuitively realizes this counter-balancing issue in the KGE type scores. The paper explicitly illustrated the issues clearly and would be nice reference for the hydrologic modelers. I think the paper is also in fairly good in shape in terms of the presentations and writing, and I don't find any major comments, and only several minor comments.**

Thank you very much for your thoughtful comment and for taking the time to carefully reading the manuscript. We appreciate that you find this work relevant to the hydrological modelling community.

**Another thought: the overall results are mostly due to the fact that the skill scores use bias, instead of the error in the magnitude (e.g., root-mean square of error, absolute error). I wonder if it is worth trying modifying KGE components into two components - absolute error and correlation. I am not requesting the authors do (I don't even know this is a good idea), but reading the paper makes me think about it.**

Thank you for the suggestion, this is an interesting idea that could be developed in a further study. Here are some preliminary results coming from a small, quick experimentation using a kind of absolute error for the volume estimate instead of β.

$$KGE'_{abs} = 1 - \sqrt{(\gamma - 1)^2 + \beta_{abs}^2 + (r - 1)^2}$$

With $KGE'_{abs}$ the modified KGE using $\beta_{abs}$, $\gamma$ the ratio between the coefficient of variation of simulated values and the coefficient of variation of observed values, and $r$ the Pearson correlation coefficient. $\beta_{abs}$ corresponds to the ratio of absolute errors to the sum of all observations, and is calculated as follows:

$$\beta_{abs} = \frac{\sum |x_s(t) - x_o(t)|}{\sum x_0(t)}$$

With $x_s(t)$ and $x_o(t)$ the simulated and observed values of a variable $x$ at a specific time step $t$. For $\beta_{abs}$, 0 would thus correspond to a perfect fit.

For synthetic models (Sect. 3 of the article) and the real case study (Sect. 4 of the article), it gives interesting results which correspond to what is expected from the visual assessment.

For the synthetic model – better score is in bold:

| Model | Bad-Bad | Bad-Good |
|---|---|---|
| $\beta$ | **0.98** | 0.88 |
| $\beta_{abs}$ | 0.22 | **0.12** |
| KGE' | **0.94** | 0.87 |
| KGE'$_{abs}$ | 0.77 | **0.87** |
| NSE | 0.92 | **0.95** |

And also, for the real case study – better score is in bold:

| Model | ANN | Bucket-type |
|---|---|---|
| $\beta$ | 0.92 | **1.00** |
| $\beta_{abs}$ | **0.27** | 0.31 |
| KGE' | 0.82 | **0.87** |
| KGE'$_{abs}$ | **0.69** | 0.66 |
| NSE | **0.87** | 0.82 |

In the future, we will consider to do an extensive study on whether this approach could be relevant for the calibration and evaluation of the performance of hydrological models.

**Minor comments**

**I do understand why NSE is included as "recommended skill scores" given the context of this paper, but still not sure if it is good idea to state so because NSE has one separate issue (underestimating variability so, peak-flow is underestimated and low flow is overestimated). I would suggest stating NSE is less impacted by counter-balance error in the hydrograph, but has its own issue for the practical applications.**

Thank you for the relevant suggestion. We modified the text to remove the ambiguity between the recommended skill scores and the skill scores that are less impacted by counterbalancing errors. As you mentioned, NSE is less impacted by counterbalancing errors but it does not seem appropriate to recommend it because of its known issue for practical applications.

L318: *"However, they have other drawbacks that are not associated with counterbalancing errors, especially the NSE with its limitation related to variability (Gupta et al., 2009)."*

L319: *"We thus recommend using performance criteria that are not or less prone to counterbalancing errors ($d_r$, KGE', KGE$_{NP}$, DE), preferably [...]"*

L22: *"We recommend using (i) performance criteria that are not or less prone to counterbalancing errors ($d_r$, modified KGE, non-parametric KGE, Diagnostic Efficiency) in a [...]"*

**Section 4. In real case study, the paper use "reservoir model" for actually some bucket type, conceptual hydrologic model. I suggest avoiding using "reservoir model" because some readers (including me) are confused with reservoir "operation" model (i.e., lake model).**

Indeed, the term "reservoir model" can be confusing for some readers. Thank you for the nice suggestion. "reservoir model" was replaced with "bucket-type model" in both text and figures.

**L264. The paper said "third flood event (May 2017) is better simulated by the ANN". I don't see this. Both ANN and reservoir models similarly underestimate the flows. Also, the statements after because are unclear to me.**

We agree that the statement of the ANN simulation being better is inappropriate for so little of a difference between the two models, which both have a poor performance on this flood event. However, we can state that the ANN simulation is *slightly* better on the timing of the flood peak ($r = 0.88$) and

the overall volumes ($\beta = 0.87$), while the bucket-type model has a better recession coefficient and variability ($\gamma = 0.99$) – see the figure and table below for the detailed analysis of the flood event. We modified the manuscript accordingly.

L263: *"The third flood event (May 2017) is poorly simulated by the models – both underestimate the flood peak – but the ANN model is more accurate in terms of timing and volume estimate, while the bucket-type model has a better recession coefficient and flow variability."*

[Figure]

| Model | NSE | KGE' | $\gamma$ | $\beta$ | r | KGE$_{NP}$ |
|---|---|---|---|---|---|---|
| ANN | 0.59 | 0.51 | 0.55 | 0.87 | 0.88 | 0.79 |
| Bucket-type | 0.35 | 0.49 | 0.99 | 0.56 | 0.73 | 0.56 |

**L269-273. I suggest using the dates to point which events are referred to.**

Thank you for the suggestion. We modified the paragraph so that the dates directly refer to the events, L268: *"The small flood events are better simulated by the ANN model than the bucket-type model: (i) the ANN model simulates them satisfactorily, except for the second one (mid-April), where the simulated discharges are overestimated; (ii) the bucket-type model does not simulate the first two events at all (mid-January and mid-April) and largely overestimates the last two (early and late June), in addition to timing errors."*

---

## Author Response (AR1)

**Manuscript hess-2022-380 – Responses to Reviewers**

We thank the referees for their careful reading and helpful comments. Our reply is given below. The page and line numbers (in the "modification to manuscript" sections) correspond to the modifications done on the revised manuscript with changes marked.

**Reviewer 1**

**1.1    Quoting: "I wonder if it would be useful to mention some of the benchmarking studies in hydrological modelling (e.g. Seibert et al. 2018) which I feel are a useful addition with regards to the performance criteria of the models."**

> **Response.** Thank you for pointing out this interesting publication. We agree that it is important to mention the possible use of more relevant benchmarks in hydrological modelling.
>
> **Modification to manuscript.**
>
> - **Page 2. Line 31.** The sentence was changed into: "Improvements can be made by working on (i) input data, [...], (iv) model calibration (Beven, 2019), and also (v) appropriate benchmarks for assessing model performance (Seibert et al., 2018)."
> - **Page 2. Line 45.** A sentence was added: "In relation to the assessment of model performance, Seibert et al. (2018) argued that the current benchmarks poorly reflect what could and should be expected of a model. They suggested to define lower and upper benchmarks based on the performance of a simple bucket-type model with few parameters, using the same data set."
> - The reference was added: "Seibert, J., Vis, M. J. P., Lewis, E., and van Meerveld, H. j.: Upper and lower benchmarks in hydrological modelling, Hydrological Processes, 32, 1120–1125, https://doi.org/10.1002/hyp.11476, 2018."

**1.2    Quoting: "L98 (Equation 11). This is the Gupta(2009) equation. This is surely wrong as the last term should be minus not plus."**

> **Modification to manuscript.** The equation was changed into: "NSE = $2\alpha r - \alpha^2 - \beta_n^2$".

**1.3    Quoting: "L133-135. I do not understand this bit. I can see there are 361 transformation between -0.36 and 0.36 but I need not understand where the logarithmic scale comes in and how you get from here to the w values"**

> **Response.** We wanted to study counterbalancing errors on a set of homothetic transformations ranging from roughly half to twice the discharges of the reference time series, which correspond to $\omega$ values of about 0.5 to 2. A linear sampling would have been uneven between underestimated and overestimated transformations (0.5–1 have a lower sample rate than 1–2). Sampling on the log-transformed interval allows to have an even distribution of the values below and above $\omega=1$. As demonstrated in Eq. (1) and Eq. (2), the common logarithmic transformations (base 10) of 0.5 and 2 nearly equal -0.30103 and 0.30103, respectively; these values are equidistant to 0, which is the common logarithmic transformation of 1 (Eq. (3)).

$$\log_{10}(0.5) \approx -0.30103 \tag{1}$$
$$\log_{10}(2) \approx 0.30103 \tag{2}$$
$$\log_{10}(1) = 0 \tag{3}$$

We decided to take a slightly larger interval to ease the reading of the graphs, i.e. [-0.36, 0.36], sampled at a 0.002 step. The exponentiation in base 10 of the sampled values allows to get an even distribution of ω values around ω=1:

$$10^{-0.36} \approx 0.4365158 \tag{4}$$
$$10^{0.36} \approx 2.290868 \tag{5}$$
$$10^0 = 1 \tag{6}$$

As $\frac{0.36}{0.002} = 180$, there are 361 transformations in total:

- 180 transformations below the ω=1 homothety in the [-0.36; -0.002] interval, with the minimum ω for -0.36 (Eq. (4))
- 180 transformations above the ω=1 homothety in the [0.002; 0.36] interval, with the maximum ω for 0.36 (Eq. (5))
- 1 transformation corresponding to the ω=1 homothety (Eq. (6))

We modified the manuscript with a better explanation of the logarithmic sampling procedure.

**Modification to manuscript.**

- **Page 6. Line 138.** The sentence was changed into: "ω values were sampled uniformly on the log-transformed interval [-0.36, 0.36] at a defined step of 0.002 to ensure a fair distribution between underestimated and overestimated transformations."
- **Page 6. Line 140.** The sentence was changed into: "The exponentiation in base 10 of the sampled values results in 361 $\omega$ values evenly distributed around the $\omega = 1$ homothety, which corresponds to the reference time series (i.e. absence of transformation)."
- **Page 6. Line 142.** The sentence was changed into: "We defined $\omega$ bounds such that the transformed peak discharge roughly ranges from half ($\omega \approx 0.437 \approx 10^{-0.36}$) to twice ($\omega \approx 2.291 \approx 10^{0.36}$) compared to the references time series."

**1.4 Quoting: "L195 (Figure 4). Would it be useful to also show the "Bad-Bad" model on this figure?"**

**Response.** Thank you, this is a great suggestion. It can be useful because it would show that, for some criteria, the "Bad-Bad" model is not even the most affected by counterbalancing errors.

**Modification to manuscript. Figure 4.** The point corresponding to the "Bad-Bad" model in Figure 2 was added to the figure.

**1.5 Quoting: "Change "consisting in" to "consisting of""**

**Modification to manuscript. Page 11. Line 215.** The sentence was changed into: "This is likely due to the nature of the equation consisting of 3 parameters [...]"

**1.6 Quoting: "Maybe change "both ways" to "both sides" or "both directions""**

**Modification to manuscript. Page 12. Line 219.** The sentence was changed into: "[...] show similar envelopes with a break point near the maximum transformation score in both directions around $\omega_1 = 1$."

**1.7 Quoting: "Change "succeed to reproduce" to "succeed in reproducing" or "successfully reproduce""**

**Modification to manuscript. Page 15. Line 271.** The sentence was changed into: "The models have overall good dynamics and successfully reproduce the observed discharges."

**1.8 Quoting: "L273-L274. "In general, the ANN model can be described as better because it is closer to the observed values in the high and low flow periods". As a hydrological modeller I agree the ANN model is better. But surely the whole point of performance criteria is to objectively decide which model is better. So how do you decide it is better when the performance criteria do not agree? There is no easy answer but I feel it is an important question that should be considered in more detail."**

**Response.** You are questioning a valid point and we agree that it deserves a better and more detailed explanation. Thank you for considering the fact that the answer is not easy. We added a paragraph in the manuscript to better explain how the ANN model can be considered as better without using performance metrics (i.e. from a subjective assessment). We also improved the description of the models for the first and second flood events.

**Modification to manuscript.**

- **Page 15. Line 273.** The sentence was changed into: "The first flood event (February 2017) is slightly underestimated by the ANN model and highly overestimated by the bucket-type model. The second flood event (March 2017) is similarly underestimated by both models but the bucket-type model demonstrates a slightly better performance."
- **Page 15. Line 290.** The sentence was moved two lines earlier: "Some events are not well simulated by both models (e.g. the May 2017 flood), which may be due to uncertainties in the input data."
- **Page 15. Line 294.** A paragraph was added: "While this statement cannot be supported by performance metrics, we believe that an expert assessment based on intuition and experience is still valuable despite being intrinsically subjective. In this particular case, one can assess the main, distinctive flaws of each model: (i) the ANN model has continuous oscillations – especially on recession and low flow periods – and lacks of accuracy during recession periods; (ii) the bucket-type model highly overestimates several flood events and is inaccurate during a lot of recession and low flow periods. Figure 6b also shows that the bucket-type model has an overall higher bias than the ANN model. Hydrological models are generally used for (i) the prediction/forecast of water flood/inrush, (ii) the management of water resources, (iii) the characterisation of a hydrosystem, and more recently (iv) the study of the impact of climate change on water resources. Most studies thus put the emphasis on volumes, and also extremes events (i.e. dry and flood periods), which in this case are more satisfactorily reproduced by the ANN model – in terms of volume estimate, timing and variability."

**1.9 Quoting: "There is no reference to Figure 7, should it be here."**

**Modification to manuscript.**

- **Page 17. Line 311.** The sentence was changed into: "The visual assessment is confirmed only by a few performance criteria: the NSE, $d_1$ and $KGE_{NP}$ (Fig. 7a)."
- **Page 17. Line 315.** The sentence was changed into: "Looking at the values of the equations' parameters (Fig. 7b), we find that [...]"

**1.10   Quoting: "L300-L303. I do not think this bit adds anything. I would remove these lines."**

**Modification to manuscript.**

- The following sentences were removed from the text: "Figure 6b shows that there is a consistent greater or equal overestimation of the reservoir model compared to the ANN model, except for the May-June period where the difference is small and insignificant compared to the February or September events. The underestimated values are similar for both approaches, except when the reservoir model overestimates the flooding events."
- **Page 18. Line 337.** The sentence was changed into: "Interestingly, the cumulative sum of the absolute bias error between simulated and observed values (Fig. 6b) is smaller for [...]"

**1.11   Quoting: "In Equation 22 the order of the parameters is alpha, beta, r. On Line 344 and subsequent lines it is r, alpha, beta. This is confusing. So when you look at (1-2-2) and you look at equation 22 everything needs to be swapped around as the 1 corresponds to r which is the last term in the equation"**

**Response.** Indeed, this is confusing. We changed the order of the y-axis number to be the same of the order of the equation, i.e. $\alpha$, $\beta$, $r$.

**Modification to manuscript.**

- **Figure 8.** The y-axis number were changed to the following order: variability, bias, timing ($\alpha$-$\beta$-r for the KGE and $\gamma$-$\beta$-r for the KGE').
- **Figure 8.** The caption was changed into: "The y-axis numbers correspond to the scaling factors of the variability, bias and timing parameters, with the default being 1-1-1."
- **Page 21. Lines 378–381.** The text was modified accordingly.

**1.12   Quoting: "Change "associated to" to "associated with""**

**Modification to manuscript. Lines 22, 62 and 401.** "associated to" was changed to "associated with".

**1.13   Quoting: "Change "include to" "include the""**

**Modification to manuscript. Page 23. Line 404.** The sentence was changed into: "Recommendations also include the use of scaling factors to emphasise [...]"

**Reviewer 2**

**2.1    Quoting: "I do understand why NSE is included as "recommended skill scores" given the context of this paper, but still not sure if it is good idea to state so because NSE has one separate issue (underestimating variability so, peak-flow is underestimated and low flow is overestimated). I would suggest stating NSE is less impacted by counterbalance error in the hydrograph, but has its own issue for the practical applications."**

> **Response.** Thank you for the relevant suggestion. We modified the text to remove the ambiguity between the recommended skill scores and the skill scores that are less impacted by counterbalancing errors. As you mentioned, NSE is less impacted by counterbalancing errors but it does not seem appropriate to recommend it because of its known issue for practical applications.
>
> **Modification to manuscript.**
>
> - **Page 19. Line 352.** The sentence was changed into: "However, they have other drawbacks that are not associated with counterbalancing errors, especially the NSE with its limitation related to variability (Gupta et al., 2009)."
> - **Page 19. Line 354.** The sentence was changed into: "We thus recommend using performance criteria that are not or less prone to counterbalancing errors ($d_1$, KGE', $KGE_{NP}$, DE)."
> - **Abstract.** The sentence was changed into: "We recommend using (i) performance criteria that are not or less prone to counterbalancing errors ($d_1$, modified KGE, non-parametric KGE, Diagnostic Efficiency), and/or [...]"

**2.2    Quoting: "Section 4.  In real case study, the paper use "reservoir model" for actually some bucket type, conceptual hydrologic model. I suggest avoiding using "reservoir model" because some readers (including me) are confused with reservoir "operation" model (i.e., lake model)."**

> **Response.** Indeed, the term "reservoir model" can be confusing for some readers. Thank you for the nice suggestion.
>
> **Modification to manuscript.**
>
> - "reservoir model" was replaced with "bucket-type model" in both text and figures.
> - "reservoir" was replaced with "bucket" in the text.

**2.3    Quoting: "L264. The paper said "third flood event (May 2017) is better simulated by the ANN". I don't see this. Both ANN and reservoir models similarly underestimate the flows.  Also, the statements after because are unclear to me."**

> **Response.** We agree that the statement of the ANN simulation being better is inappropriate for so little of a difference between the two models, which both have a poor performance on this flood event. However, we can state that the ANN simulation is *slightly* better on the timing of the flood peak ($r = 0.88$) and the overall volumes ($\beta = 0.87$), while the bucket-type model has a better recession coefficient and variability ($\gamma = 0.99$) – see the figure and table below. We modified the manuscript accordingly.

[Figure]

| Model | NSE | KGE' | γ | β | r | KGE$_{NP}$ |
|---|---|---|---|---|---|---|
| ANN | 0.59 | 0.51 | 0.55 | 0.87 | 0.88 | 0.79 |
| Bucket-type | 0.35 | 0.49 | 0.99 | 0.56 | 0.73 | 0.56 |

**Modification to manuscript.**

- **Page 15. Line 277.** The sentence was changed into: "The third flood event (May 2017) is poorly simulated by the models – both underestimate the flood peak – but the ANN model is more accurate in terms of timing and volume estimate, while the bucket-type model has a better recession coefficient and flow variability."

**2.4 Quoting: "L269-273. I suggest using the dates to point which events are referred to."**

**Response.** Thank you for the suggestion. We modified the paragraph so that the dates directly refer to the events.

**Modification to manuscript. Page 15. Line 284.** The sentence was changed into: "The small flood events are better simulated by the ANN model than the bucket-type model: (i) the ANN model simulates them satisfactorily, except for the second one (mid-April), where the simulated discharges are overestimated; (ii) the bucket-type model does not simulate the first two events at all (mid-January and mid-April) and largely overestimates the last two (early and late June), in addition to timing errors."

**Community comment 1 (CC1)**

**3.1    Quoting: "In the last sentence of the abstract, the authors mention the use of multi-criteria framework in their recommendation. On the need to consider a particular "goodness-of-fit" metric within the multi-criteria framework, the authors could clarify on other specific requirements apart from the general condition that the performance criteria should be less or not prone to counterbalancing. Furthermore, the use of several criteria for a particular calibration can complicate the applications of automation of famous search strategies or algorithms (Onyutha 2022). It is upon this basis that a number of performance criteria which are not mathematically and statistically related tend to be formed into single metric. For instance, Kling-Gupta Efficience combines three components including measures of bias, variability and linear correlation between observed (X) and modelled (Y) series. Thus, the authors should provide more considered justification for their recommendation of the use of multi-criteria framework for calibration of hydrological models."**

> **Response.** Thank you for this relevant examination. Indeed, the explanation around the use of a multi-criteria framework is unclear. All things considered, it seems that the recommendation of using a multi-criteria framework is not appropriate in this article, as its scope is to identify and raise awareness about the problem of counterbalancing errors. The mention of the multi-criteria framework, its benefit and also relevant references are already mentioned is the introduction, so we changed the recommendations to be consistent with the findings of this study.
>
> **Modification to manuscript.**
>
> - **Abstract.** The sentence was changed into: "We recommend using (i) performance criteria that are not or less prone to counterbalancing errors ($d_1$, modified KGE, non-parametric KGE, Diagnostic Effiency), and/or (ii) [...]"
> - **Page 19. Line 354.** The sentence was changed into: "We thus recommend using performance criteria that are not or less prone to counterbalancing errors ($d_1$, KGE', $KGE_{NP}$, DE)."
> - **Page 21. Line 384.** The sentence was changed into: "This is why a multi-criteria framework can strengthen the evaluation of models and reduce the uncertainty associated with the interpretation of individual performance criteria scores."

**3.2    Quoting: "Most of (if not all) the metrics used in this study rely on the assumption that X and Y are linearly related. Note that X and Y can be so highly dependent yet it may be nearly impossible to detect the dependence using classical dependence metric (Székely et al. 2007). In other words, the authors should clarify on whether the model performance results of this study may not have been affected by the said assumption.**

> **Response.** Thank you for pointing out this implicit assumption of the KGE and its variants. Although this study focuses on counterbalancing errors in widely used performance criteria and not (so much) on the correlation between X and Y, it is important to clarify whether the data linearity between X and Y is skewed – which is often the case in hydrological modelling – to better appreciate the model performance results. Note that we also included the non-parametric KGE (Pool et al., 2018), which is based on the Spearman correlation coefficient and the flow duration curve, and has no assumption of data linearity.
>
> **Modification to manuscript.**

- **Page 6. Line 144.** The sentence was changed into: "Note that (i) the data linearity between simulated and observed values is verified, and (ii) ω homotheties still induces [...]"
- **Page 15. Line 292.** A sentence was added: "Also, the data linearity between simulated and observed values is slightly skewed for both models, which can affect the relevance of *r* (Barber et al., 2020)."

**3.3    Quoting: "Most of the performance criteria (especially Nash Sutcliffe Efficiency NSE (Nash and Sutcliffe, 1970) and its variants) comprise some forms of the well-known coefficient of determination (R-squared) (see Onyutha, 2022). R-squared is known to have various short comings. To address these short comings, new metrics including the revised R-squared (RRS) and hydrological model skill score E (Onyutha 2022) were developed. Thus, instead of focussing on NSE and its variants, the authors should compare results of many other performance criteria such as RRS and E. Accordingly, Figure 7 and Table 1 in this manuscript can be updated. The MATLAB codes to compute RRS and E can be downloaded via https://doi.org/10.5281/zenodo.6570905 and the codes can also be found as supplementary material to the paper by Onyutha (2022)"**

> **Response.** Thank you for the suggestion of these two innovative performance criteria and the associated code. The results for the synthetic models and the real case study have been added as a figure in appendix alongside other commonly used performance criteria (RMSE, $R^2$, d, $d_r$).

> **Modification to manuscript.**

> - **Appendix A.** An appendix was added: "Appendix A: Common and recently developed performance criteria applied to the synthetic time series and the real case study"
> - **Figure A1.** A figure was added in Appendix A with the following caption: "Figure A1: Score of the BB and BG transformations according to other common and recently developed performance criteria: the Root Mean Square Error (RMSE), the coefficient of determination $R^2$, the index of agreement d (Willmott, 1981), the refined index of agreement $d_r$ (Willmott et al., 2012), the Onyutha efficiency E and the revised R-squared RRS (Onyutha, 2022).
> - **Figure A2.** A figure was added in Appendix A with the following caption: "Figure A2: Score of the ANN and bucket-type models according to other common and recently developed performance criteria: the Root Mean Square Error (RMSE), the coefficient of determination $R^2$, the index of agreement d (Willmott, 1981), the refined index of agreement $d_r$ (Willmott et al., 2012), the Onyutha efficiency E and the revised R-squared RRS (Onyutha, 2022).
> - **Page 7. Line 162.** A sentence was added: "Further results for common and recently developed performance criteria are presented in Fig. A1."
> - **Page 17. Line 314.** A sentence was added: "Further results for common and recently developed performance criteria are presented in Fig. A2."
> - **Page 17. Line 315.** The sentence was changed into: "It is interesting to note how similar these results are to those of the synthetic example (Fig. 3a, Fig. A1)."

**3.4    Quoting: "According to Legates & McCabe (2013), the refinement of Index of Agreement (IOA) (Willmott, 1981) made by Willmott et al. (2012) especially regarding the extension of the IOA bound from 1 to 0 was unnecessary. Check Legates & McCabe (2013) for other limitations of the refined IOA. Therefore, could the authors make use of the original form of IOA for their model performance evaluation and analyses?"**

**Response.** Thank you for pointing this interesting discussion about the refined index of agreement. Following your suggestion, we changed the refined index of agreement to the modified index of agreement (Willmott et al., 1985). We chose to use the modified index of agreement because it is less sensitive to outlier, which is relevant in our case study.

**Modification to manuscript.**

- **Page 3. Line 67.** The sentence was changed into: "[...] as well as more traditional criteria such as the NSE or the modified index of agreement ($d_1$) for comparison purpose."
- **Page 5. Line 124.** The sentence was changed into: "Willmott et al. (1985) proposed a modified index of agreement, which aim to address the issues associated with $r$ and the coefficient of determination, as well as the sensitivity of the original index of agreement to outliers (Legates and McCabe Jr., 1999):"
- Equation 20 was changed to the one of the modified index of agreement.
- **Abstract.** The sentence was changed into: "[...] as well as the Nash-Sutcliffe Efficiency (NSE) and the modified index of agreement ($d_1$) [...]"
- **Abstract.** The sentence was changed into: "We recommend using (i) performance criteria that are not or less prone to counterbalancing errors ($d_1$, modified KGE, non-parametric KGE, Diagnostic Efficiency) [...]"
- "$d_r$" was replaced with "$d_1$" throughout the manuscript.
- Figure 3, 4, 5 and 7 were updated accordingly.
- **Table 1.** The advantages of the modified index of agreement were changed into: "Address the shortcomings of $r$ and the coefficient of determination (Willmott et al., 1981)" and "The score is less sensitive to errors concentrated in outliers in comparison to the original index of agreement (Willmott et al., 1985)"

**Additional modification to the manuscript**

**4.1    Grammar and error check**

**Modification to manuscript.**

- **Page 5. Line 109.** As the FDC abbreviation is already defined Line 84, the sentence was changed into: "The non-parametric form of the variability is calculated using the FDC [...]"
- Remove repetition of *the*, e.g. "The NSE and  $d_1$".
  - Page 3. Line 66.
  - Page 7. Line 162.
  - Page 9. Line 188.
  - Page 9. Line 189.
  - Page 11. Line 209.
  - Page 12. Line 219.
  - Figure 5. Caption.
  - Page 13. Line 230.
- The reference of a Python library was corrected: "[...] Bayesian Optimization (Nogueira, 2014) [...]"
- **Page 17. Line 312.** Add missing "of": "However, the KGE and most of its variants [...]"

**References**

Barber, C., Lamontagne, J. R., and Vogel, R. M.: Improved estimators of correlation and $R^2$ for skewed hydrologic data, Hydrological Sciences Journal, 65, 87–101, https://doi.org/10.1080/02626667.2019.1686639, 2020.

Onyutha, C.: A hydrological model skill score and revised R-squared, Hydrology Research, 53, 51–64, https://doi.org/10.2166/nh.2021.071, 2022.

Seibert, J., Vis, M. J. P., Lewis, E., and van Meerveld, H. j.: Upper and lower benchmarks in hydrological modelling, Hydrological Processes, 32, 1120–1125, https://doi.org/10.1002/hyp.11476, 2018.

Willmott, C. J.: On the validations of models, Phys. Geogr., 2, 184–194, https://doi.org/10.1080/02723646.1981.10642213, 1981.

Willmott, C. J., Ackleson, S. G., Davis, R. E., Feddema, J. J., Klink, K. M., Legates, D. R., O'Donnell, J., and Rowe, C. M.: Statistics for the evaluation and comparison of models, J. Geophys. Res., 90, 8995, https://doi.org/10.1029/JC090iC05p08995, 1985.

Willmott, C. J., Robeson, S. M., and Matsuura, K.: A refined index of model performance, Intern. J. Climatol., 32, 2088–2094, https://doi.org/10.1002/joc.2419, 2012.